# Dreamweaver: Learning Compositional World Models from Pixels

**Junyeob Baek**[*]
KAIST

**Yi-Fu Wu**
Rutgers University

**Gautam Singh**
Rutgers University

**Sungjin Ahn**[*]
KAIST,
New York University

arXiv:2501.14174v5 [cs.CV] 10 Apr 2025

## Abstract

Humans have an innate ability to decompose their perceptions of the world into objects and their attributes, such as colors, shapes, and movement patterns. This cognitive process enables us to imagine novel futures by recombining familiar concepts. However, replicating this ability in artificial intelligence systems has proven challenging, particularly when it comes to modeling videos into compositional concepts and generating unseen, recomposed futures without relying on auxiliary data, such as text, masks, or bounding boxes. In this paper, we propose **Dreamweaver**, a neural architecture designed to discover hierarchical and compositional representations from raw videos and generate compositional future simulations. Our approach leverages a novel Recurrent Block-Slot Unit (RBSU) to decompose videos into their constituent objects and attributes. In addition, Dreamweaver uses a multi-future-frame prediction objective to capture disentangled representations for dynamic concepts more effectively as well as static concepts. In experiments, we demonstrate our model outperforms current state-of-the-art baselines for world modeling when evaluated under the DCI framework across multiple datasets. Furthermore, we show how the modularized concept representations of our model enable compositional imagination, allowing the generation of novel videos by recombining attributes from previously seen objects. [cun-bjy.github.io/dreamweaver-website](cun-bjy.github.io/dreamweaver-website)

## 1 Introduction

The primary function of the brain is believed to be the construction of an internal model of the world from sensory inputs like vision—a concept often referred to as *world models* (Ha & Schmidhuber, 2018). This construction involves developing two fundamental processes: *knowing* and *thinking* (Summerfield, 2022; Lake et al., 2017; Fodor et al., 1975). The knowing function entails how to compositionally structure and encode knowledge from experiences through representation learning. The thinking function enables the utilization of this encoded knowledge for abilities such as reasoning, planning, imagining, and causal inference (Goyal & Bengio, 2022; Schölkopf et al., 2021a). Due to its generative and inferential nature, the thinking function necessitates some form of generative learning (Parr & Friston, 2018; Kurth-Nelson et al., 2023; Schwartenbeck et al., 2023).

A key aspect distinguishing the world models in humans and AI currently is *compositionality* (Smolensky et al., 2022; Schölkopf et al., 2021c; Lake et al., 2017; Goyal & Bengio, 2022; Greff et al., 2020; Behrens et al., 2018), the capability to understand or construct complex concepts as a composition of simpler concepts. Recent studies in neuroscience suggest that humans can understand and adapt to various novel situations because the brain supports compositional representation and generation (Kurth-Nelson et al., 2023; Schwartenbeck et al., 2023; Behrens et al., 2018; Bakermans et al., 2023). However, current world models in AI are limited in this ability for the following reasons.

Most of compositionality in AI is currently approached via language-conditioned image/video generation models such as DALL·E (Ramesh et al., 2021; 2022) and Sora (Cho et al., 2024; Millière). These models do offer a degree of compositionality (e.g., generating an image of an avocado chair) through language as a medium, which is convenient for human interaction. However, achieving

---

[*]Correspondence to `wnsdlqjtm@kaist.ac.kr` and `sungjin.ahn@kaist.ac.kr`.

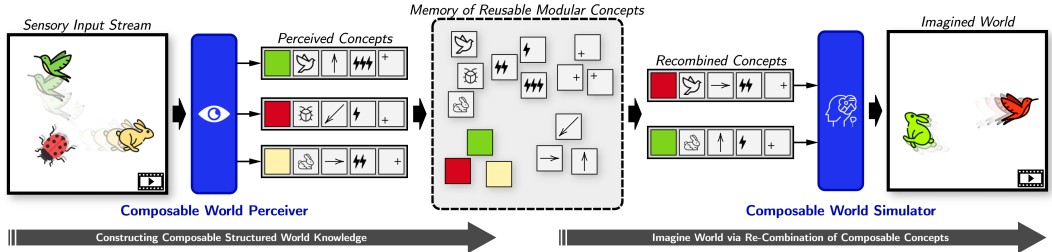

**Figure 1: Overview of the Dreamweaver Framework.** Our aim is to take a sequential unstructured sensory stream and bind the low-level information into abstract modular concepts to build a memory of reusable concepts, called concept library—all without text and in an unsupervised way. These concepts include both static factors such as color and shape as well as dynamic factors such as direction and speed of motion. Finally, we seek to recombine these concepts, e.g., in a novel configuration, and imagine an unseen world.

compositionality more broadly, without relying on language, remains a challenge. This is crucial because there is a vast amount of world knowledge that is difficult to accurately articulate through language. Another limitation of this language-dependent approach is that it addresses only half of the problem. This is because the most challenging part of the knowing problem, namely discovering a composable knowledge representation from unstructured raw sensory inputs, is sidestepped as the core structure is *provided* through the token-compositional structure inherent in language, rather than being discovered. To the best of our knowledge, this crucial ability of extracting composable world knowledge *from video without language* and enabling compositional imagination, has yet to be realized in machine learning.

In this paper, we take the first step toward this challenge. Specifically, we focus on *unsupervised learning of compositional world models from image sequences in a way to support both the compositional knowing and thinking processes*. We approach this challenge from the perspective of Object-Centric Learning (OCL) (Greff et al., 2020). OCL aims to learn to develop a compositional and modular representation from visual observations in an unsupervised way. However, existing OCL approaches lack some key properties required for a truly compositional world model. For instance, most OCL methods, such as Slot Attention (SA) (Locatello et al., 2020) and its subsequent models (Singh et al., 2022a; Kipf et al., 2021; Singh et al., 2022b; Seitzer et al., 2022) maintain a monolithic, entangled representation for representing an object while lacking a compositional structure within an object. This issue has recently been addressed by SysBinder (Singh et al., 2023; Wu et al., 2024) by introducing a block-slot representation that offers a nested compositional structure, referred to as blocks, within an object slot. These works (Singh et al., 2023; Wu et al., 2024) showed that this block-slot representation enables the generation of novel scene imaginations by recombining static blocks concepts in an out-of-distribution manner.

However, since these models are designed for static images, it is unclear whether they can be extended for dynamic sequence modeling and serve as a foundation for image-based compositional world models. A key uncertainty and challenge is whether dynamic concept blocks, such as direction (e.g., 'to the right') and speed (e.g., 'fast' or 'slow'), can emerge solely from observational learning. Our key hypothesis is that object-centric representation learning alone is insufficient. Instead, both a block-slot representation and a temporally predictive objective function are necessary—components that were not utilized in previous works. If successful, this proof-of-concept approach would represent a step toward the grand challenge of generating novel videos without relying on language, through the composition of both static and dynamic block-slot concepts, such as a "flying green rabbit," as shown in Fig 1.

We achieve this goal by introducing a neural architecture named *Dreamweaver*. Dreamweaver encodes a temporal context window of $T$ past images using a novel Recurrent Block-Slot Unit (RBSU). The RBSU represents its state as a set of slot states, each updated independently. This is then passed through a block-slot bottleneck, mapping the potentially entangled monolithic slot state into a composition of independent blocks. Since a block vector can only obtain its value through attention to the prototype concept library, this process can also be seen as implementing an attraction process at each step. At the end of the temporal encoding, Dreamweaver predicts the observation at a future time step from the latest block-slot representation. We found that this predictive reconstruction objective is indeed crucial in making the dynamic concept abstractions emerge. In experiments, Dreamweaver outperforms state-of-the-art object-centric methods under the DCI framework across several datasets.

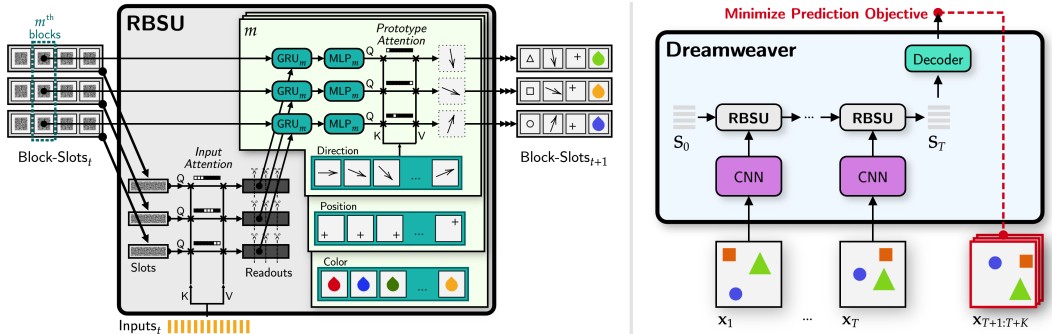

**Figure 2: Model Architecture.** *Left:* The Recurrent Block-Slot Unit (RBSU) is a recurrent unit designed for processing sequences where each item is a set of vectors. RBSU maintains and updates Block-Slots, which represent compositional and semantic concepts such as shape, color, and motion direction. *Right:* The Dreamweaver model encodes video inputs into Block-Slot representations, which pass through a series of RBSUs with a recurrent structure. It then predicts future frames by decoding the extracted Block-Slots using a transformer decoder, training to minimize the predictive objective.

Moreover, we demonstrate how RBSU's modularized concept representations enable compositional imagination, generating novel videos that combine attributes from previously seen objects.

The main contributions of the paper are as follows: We introduce the Dreamweaver model for compositional world modeling from pixels. We introduce a novel recurrent module, named Recurrent Block-Slot Units. Our model is the first in object-centric learning that can learn both static and dynamic composable concepts in an unsupervised way. By contrast, previous works were modeling only static images and able to learn only static concepts. We also found that using a predictive imagination loss is crucial in achieving this, in addition to the architectural inductive bias of block-slot representation. We found that previous temporal object-centric models cannot develop abstraction of dynamic concepts due to their autoencoding objective. Finally, for the first time in this area, we demonstrate the ability of Dreamweaver to simulate a future from an out-of-distribution configuration of discovered concepts. While our model shares a common limitation of current state-of-the-art object-centric learning models—it is not yet applicable to highly complex scene images—we believe that the success of this proof-of-concept represents a significant step toward the grand challenge of composing novel videos without relying on language.

## 2 RECURRENT BLOCK-SLOT UNITS

A *Recurrent Block-Slot Unit* or *RBSU* is a general-purpose recurrent unit for sequence modeling that we propose in this work. Given an input sequence $\mathbf{E}_1, \ldots, \mathbf{E}_T$, where each item in the sequence $\mathbf{E}_t \in \mathbb{R}^{L \times D_{\text{input}}}$ is a collection of $L$ vectors, RBSU works by processing the sequence recurrently. It starts with an initial state representation $\mathbf{S}_0$, and for each item $\mathbf{E}_t$ in the input sequence, applies the RBSU to update the state from $\mathbf{S}_{t-1}$ to $\mathbf{S}_t$:

$$\mathbf{S}_0 \leftarrow \text{Initialize}() \qquad \Longrightarrow \qquad \mathbf{S}_t \leftarrow \text{RBSU}(\mathbf{E}_t, \mathbf{S}_{t-1}).$$

Importantly, RBSU maintains a disentangled state representation $\mathbf{S}_t$ called the *block-slot representation* or simply *block-slots*. The block-slot representation is a collection of $N$ vectors called *slots* i.e., $\mathbf{S}_t \in \mathbb{R}^{N \times MD}$. We denote each $n$-th slot in $\mathbf{S}_t$ as $\mathbf{s}_{t,n} \in \mathbb{R}^{MD}$. Furthermore, each slot is internally disentangled and constructed by concatenating $M$ vectors of size $D$ called *blocks*. We denote the $m$-th block within a slot $\mathbf{s}_{t,n}$ as $\mathbf{s}_{t,n,m} \in \mathbb{R}^D$. For modeling multi-object video inputs, a slot $\mathbf{s}_{t,n}$ can represent an object while a block $\mathbf{s}_{t,n,m}$ within the slot can represent an intra-object reusable concept *e.g.*, color, shape, or direction of motion.

### 2.1 BOTTOM-UP ATTENTION

In the first step in an RBSU, the $N$ slots of the previous time-step $\mathbf{S}_{t-1}$ act as queries and attend over the current $L$ input features $\mathbf{E}_t$ to obtain $N$ bottom-up *readout* vectors $\mathbf{U}_t \in \mathbb{R}^{N \times MD}$. Following Singh et al. (2023); Locatello et al. (2020), this is performed via inverted attention and renormalization

(Wu et al., 2023a) as follows:

$$\mathbf{A}_t = \underset{N}{\mathrm{softmax}} \left( \frac{q(\mathbf{S}_{t-1}) \cdot k(\mathbf{E}_t)^T}{\sqrt{MD}} \right) \implies \mathbf{A}_{t,n,l} = \frac{\mathbf{A}_{t,n,l}}{\sum_{l=1}^{L} \mathbf{A}_{t,n,l}} \implies \mathbf{U}_t = \mathbf{A}_t \cdot v(\mathbf{E}_t),$$

Next, in the readout $\mathbf{U}_t$, we split each $n$-th row vector $\mathbf{u}_{t,n} \in \mathbb{R}^{MD}$ into $M$ equal *readout chunks*, where each $m$-th chunk is denoted by $\mathbf{u}_{t,n,m}$. The chunks shall be used to update their corresponding block in the block-slot state representation.

## 2.2 INDEPENDENT BLOCK ATTRACTOR DYNAMICS

The next step in an RBSU is to update each block in the current block-slot state representation via per-block independent recurrent modules. Importantly, we incorporate attractor dynamics within these recurrent modules to facilitate the convergence of each block's state to a reusable representation. We perform the following per-block operations:

**GRU and MLP.** First, each previous block state $\mathbf{s}_{t-1,n,m}$ is independently updated using its corresponding readout chunk $\mathbf{u}_{t,n,m}$ by applying a GRU to incorporate the bottom-up information. The resulting block is then fed to an MLP with a residual connection as follows:

$$\tilde{\mathbf{s}}_{t,n,m} = \mathrm{GRU}_{\phi_m}(\mathbf{s}_{t-1,n,m}, \mathbf{u}_{t,n,m}) \implies \bar{\mathbf{s}}_{t,n,m} = \tilde{\mathbf{s}}_{t,n,m} + \mathrm{MLP}_{\phi_m}(\mathrm{LN}(\tilde{\mathbf{s}}_{t,n,m})).$$

We maintain separate GRU and MLP weights for each $m$ to encourage modularity between blocks. Here, the $\tilde{\mathbf{s}}_{t,n,m}$ and $\bar{\mathbf{s}}_{t,n,m}$ denote the intermediate block states after applying GRU and the residual MLP, respectively.

**Block-Slot Attractor Dynamics.** Since GRU and MLP alone are found insufficient for making a block's state reach a reusable attractor state (Singh et al., 2023), we explicitly incorporate a learned memory of prototypes and make each block perform dot-product attention over this learned memory to retrieve a state as follows:

$$\hat{\mathbf{s}}_{t,n,m} = \left[ \underset{N_{\mathrm{prototypes}}}{\mathrm{softmax}} \left( \frac{\bar{\mathbf{s}}_{t,n,m} \cdot \mathbf{C}_m^T}{\sqrt{d}} \right) \right] \cdot \mathbf{C}_m.$$

To maintain modularized representations from GRU and MLP, we utilize a separate library of prototypes $\mathbf{C}_m \in \mathbb{R}^{N_{\mathrm{prototypes}} \times D}$ for each $m$. Notably, each library is initialized with $N_{\mathrm{prototypes}}$ learnable vectors, which start with random values and are subsequently learned through backpropagation to capture emergent semantic concepts.

## 2.3 BLOCK INTERACTION

Finally, although we maintain independent information processing pathways, for improved flexibility, it is desirable to let the blocks in $\hat{\mathbf{S}}_t$ interact: $\mathbf{S}_t = \mathrm{BlockInteraction}(\hat{\mathbf{S}}_t)$. To implement this interaction step, we first flatten the slots into a collection of $NM$ block vectors, feed them to a single-layer transformer, and then reshape the output back to the original shape $\mathbf{S}_t \in \mathbb{R}^{N \times MD}$.

# 3 DREAMWEAVER: COMPOSITIONAL WORLD MODEL VIA PREDICTIVE IMAGINATION

In this section, we propose and describe a novel compositional world model called *Dreamweaver* using the proposed RBSU. Broadly, Dreamweaver takes video frames $\mathbf{x}_1, \ldots, \mathbf{x}_T$ as input, constructs the world state in terms of composable tokens, and predicts $K$ future video frames $\hat{\mathbf{x}}_{T+1}, \ldots, \hat{\mathbf{x}}_{T+K}$:

$$\hat{\mathbf{x}}_{T+1}, \ldots, \hat{\mathbf{x}}_{T+K} = \mathrm{Dreamweaver}(\mathbf{x}_1, \ldots, \mathbf{x}_T),$$

where $T$ is a temporal context window size and the frames belong to $\mathbb{R}^{C \times H \times W}$. Dreamweaver is implemented as an encoder-decoder architecture:

$$\mathbf{S}_T = f_\phi(\mathbf{x}_1, \ldots, \mathbf{x}_T) \implies \hat{\mathbf{x}}_{T+k} = g_\theta(\mathbf{S}_T, k)$$

where the encoder $f_\phi$ encodes the video $\mathbf{x}_1, \ldots, \mathbf{x}_T$ into slots $\mathbf{S}_T$ leveraging RBSU. Given the slots $\mathbf{S}_T$ and a step indicator $k$, where $k$ is the relative temporal index of the target frame, the decoder $g_\theta$ predicts the target frame $\hat{\mathbf{x}}_{T+k}$ via an autoregressive image transformer (Chen et al., 2020; Singh et al., 2022a;b; 2023).

### 3.1 Encoding via Spatiotemporal Concept Binding

In the encoder of Dreamweaver, each input frame $\mathbf{x}_t$ is processed by a CNN to output a feature map. Next, we add positional embeddings to this feature map, flatten it, and feed it to an MLP to obtain $\mathbf{E}_t \in \mathbb{R}^{L \times D}$. This is similar to Locatello et al. (2020); Singh et al. (2022b; 2023). The sequence $\mathbf{E}_1, \ldots, \mathbf{E}_T$ is then processed by the RBSU to produce slots $\mathbf{S}_1, \ldots, \mathbf{S}_T$:

$$\mathbf{E}_t = \text{CNN}_\phi(\mathbf{x}_t) \qquad \Longrightarrow \qquad \mathbf{S}_t \leftarrow \text{RBSU}_\phi(\mathbf{E}_t, \mathbf{S}_{t-1}).$$

Here, we obtain the initial block-slot state $\mathbf{S}_0$ by sampling it from a learned Gaussian distribution. That is, for all $n = 1, \ldots, N$, we sample $\mathbf{s}_{0,n} \sim \mathcal{N}(\boldsymbol{\mu}_\phi, \boldsymbol{\sigma}_\phi)$, where $\boldsymbol{\mu}_\phi, \boldsymbol{\sigma}_\phi \in \mathbb{R}^{MD}$ are learned parameters.

### 3.2 Decoding via Autoregressive Image Transformer

The extracted slots $\mathbf{S}_T$ are decoded leveraging an autoregressive image transformer to predict the future frames $\mathbf{x}_{T+1}, \ldots, \mathbf{x}_{T+K}$. Similar to the powerful architectures used in (Singh et al., 2022a;b; 2023), our transformer decoder does not directly predict the target images in pixel space but instead predicts their discrete token representation. We employ a Discrete VAE (dVAE) (Ramesh et al., 2021; Singh et al., 2022a) to transform an image into a sequence of $L'$ integer tokens $z_{T+k,1}, \ldots, z_{T+k,L'} \in \mathcal{V}$ and vice-versa, where $\mathcal{V}$ denotes a vocabulary.

To train the transformer, we adopt the standard practice in language modeling (Vaswani et al., 2017; Singh et al., 2022a) and perform parallelized training of the autoregressive transformer via causal masking. That is, we first map each token $z_{T+k,l} \in \mathcal{V}$ to an embedding $\mathbf{e}_{T+k,l} \in \mathbb{R}^D$ by retrieving the token's embedding from a learned dictionary and adding a positional encoding as follows: $\mathbf{e}_{T+k,l} = \text{Dictionary}_\theta(z_{T+k,l}) + \mathbf{p}_{\theta,l}$. Next, we provide the embeddings $\mathbf{e}_{T+k,1}, \ldots, \mathbf{e}_{T+k,L'-1}$ to the transformer decoder that utilizes a beginning-of-sequence (BOS) token and causal masking to generate the next-token log probabilities for each input token:

$$\mathbf{o}_{T \to T+k,1}, \ldots, \mathbf{o}_{T \to T+k,L'} = \text{Transformer}_\theta(\text{BOS}_{\theta,k}, \mathbf{e}_{T+k,1}, \ldots, \mathbf{e}_{T+k,L'-1}; \text{cond} = \mathbf{S}_T),$$

Here, the notation $\mathbf{o}_{t_{\text{source}} \to t_{\text{target}}, l} \in \mathbb{R}^{|\mathcal{V}|}$ denotes the predicted log probabilies over $\mathcal{V}$ for the $l$-th token of the image at time-step $t_{\text{target}}$ given the slots inferred at time-step $t_{\text{source}}$. Additionally, the BOS token serves a dual purpose: it provides the initial input to the transformer and acts as a step indicator, representing the relative temporal index $k$. We maintain a dedicated BOS token for each $k$, denoted as $\text{BOS}_{\theta,k}$. The detailed method for the conditional prediction of the $k$-th frame is described in Appendix B.2.

### 3.3 Training Objective

The complete model is trained by minimizing a cross-entropy loss averaged over all values of $k = 1, \ldots, K$, and all token indices $l = 1, \ldots, L'$:

$$\mathcal{L}_{\text{Dreamweaver}}(\theta, \phi) = \frac{1}{KL'} \sum_{k=1}^{K} \sum_{l=1}^{L'} \text{CrossEntropy}(z_{T+k,l}, \mathbf{o}_{T \to T+k,l})$$

In practice, we train the dVAE jointly with the Dreamweaver by using a combined loss $\mathcal{L}(\theta, \phi) = \mathcal{L}_{\text{Dreamweaver}}(\theta, \phi) + \mathcal{L}_{\text{dVAE}}(\theta, \phi)$. For more details about the dVAE, see Appendix section B.1.

**Need for a Predictive Objective.** The purpose of introducing a predictive objective in contrast to a simple reconstruction objective commonly used in prior works (Kipf et al., 2021; Singh et al., 2022b; Elsayed et al., 2022) is two-fold: *i)* The predictive objective incentivizes RBSU to capture not just static factor primitives, such as *color* or *shape*, but also dynamical primitives, such as *direction* or *speed*, since the latter is necessary when performing future prediction but much less important when the goal is to merely reconstruct the present frame. *ii)* The ability to generate future frames naturally provides a world model that can be utilized to roll out long future trajectories by autoregressively feeding each generated frame back into the model.

## 4 Related Works

**Learning Object-centric Compositional Representations.** Our work is influenced by recent research in unsupervised object-centric representation learning (Locatello et al., 2020; Engelcke et al.,

2020; Burgess et al., 2019; Greff et al., 2017; 2019; 2020; Van Steenkiste et al., 2018; Zoran et al., 2021; Veerapaneni et al., 2019; Ding et al., 2020; Seitzer et al., 2022; Lowe et al., 2022; Sajjadi et al., 2022), where multi-object scenes are decomposed into entity-level representations corresponding to the objects in the scene. In particular, our method is built upon the family of Slot Attention-based models (Locatello et al., 2020; Singh et al., 2022a; Wu et al., 2023b; Jiang et al., 2023) and related to several Slot Attention for Video models (Kipf et al., 2021; Elsayed et al., 2022; Singh et al., 2022b; Bao et al., 2023; Zadaianchuk et al., 2023; Singh et al., 2024; Jiang et al., 2024). Unlike our method, these models are trained with a reconstruction objective instead of a predictive objective and generally do not support imagination of future frames. Furthermore, these slot-based models only decompose the scene into object-level representations, whereas our model discovers concept-level representations in the form of block-slot representations. While previous work (Singh et al., 2023; Wu et al., 2024) also learned static concept-level representations from images, our method is the first to be applied to videos and additionally capture dynamic concepts. Another line of research learns bounding boxes for the objects in the scene, decomposing the objects into *where* and *what* representations (Eslami et al., 2016; Crawford & Pineau, 2019b; Kosiorek et al., 2018; Lin et al., 2020b; Crawford & Pineau, 2019a; Jiang et al., 2019; Lin et al., 2020a). However, these methods also do not further decompose into concept-level representations. Lastly, several works (Wu et al., 2021; 2022) support future frame prediction by training a transformer on a set of pretrained object-representations (Lin et al., 2020b; Kipf et al., 2021; Singh et al., 2022b). Our method instead is trained end-to-end, which allows dynamic concepts to be captured in the representations and naturally reuses them to predict future frames. Additional related works are detailed in Appendix A.

## 5 EXPERIMENTS

We evaluate the following questions: *(1)* How effectively can Dreamweaver infer modular concepts—both static and dynamic—from RGB videos without any supervision? *(2)* Can Dreamweaver generate new videos by composing novel configurations of the inferred concepts? *(3)* How well does Dreamweaver's representation support out-of-distribution (OOD) generalization on downstream reasoning tasks? Finally, we conduct an ablation study to evaluate various design choices.

**Datasets.** We experiment on five datasets spanning two axes of complexity: *Dynamical Complexity* and *Visual Complexity*. Along the axis of dynamical complexity, our datasets Moving-Sprites, Moving-CLEVR, and Moving-CLEVRTex exhibit *low* dynamical complexity i.e., objects move uniformly in a specific direction throughout the video (e.g., up, down, left, right). On the other hand, our datasets Dancing-Sprites and Dancing-CLEVR exhibit *high* dynamical complexity, featuring multi-step "dance" patterns (e.g., a clockwise square dance pattern: `right → down → left → up`, as shown in Figure 14) or multi-step color-change patterns (see Figure 15). Along the axis of visual complexity, we have 3 levels: (i) The simplest are Moving-Sprites and Dancing-Sprites with 2D sprites on a black canvas. (ii) Moving-CLEVR and Dancing-CLEVR significantly increase visual complexity, with 3D scenes featuring solid-colored 3D rubber objects, realistic lighting, and shading. (iii) In line with previous work (Singh et al., 2023; Wu et al., 2024), the highest level of visual complexity is in Moving-CLEVRTex, where objects have complex textures on them.

**Baselines.** We compare our model against three unsupervised representation learning baselines: RSSM (Hafner et al., 2018), STEVE (Singh et al., 2022b), and SysBinder (Singh et al., 2023). RSSM is a widely used world modeling framework that represents a video frame via single-vector representation. STEVE is a state-of-the-art method that represents each video frame as a set of per-object latent vectors (called slots). STEVE uses Slot Attention (Locatello et al., 2020) to represent scenes in a structured manner by spatially binding objects into slot representations. Like STEVE, SysBinder also provides a slot representation of each video frame; however, it further disentangles a slot as a concatenation of several blocks—each block representing one factor e.g., color, shape, etc. Since SysBinder was originally designed for image data, it can only be applied independently per video frame and is not capable of utilizing the temporal context. Therefore, for a fair comparison with our model, we modify SysBinder for videos to obtain a stronger baseline as follows. We equip SysBinder with a recurrent encoder (Kipf et al., 2021) to handle multiple frames. We refer to this modified model as SysBinder for simplicity. For all baselines, we use the same length of conditioning frames as our model. The baselines are trained with a reconstruction objective.

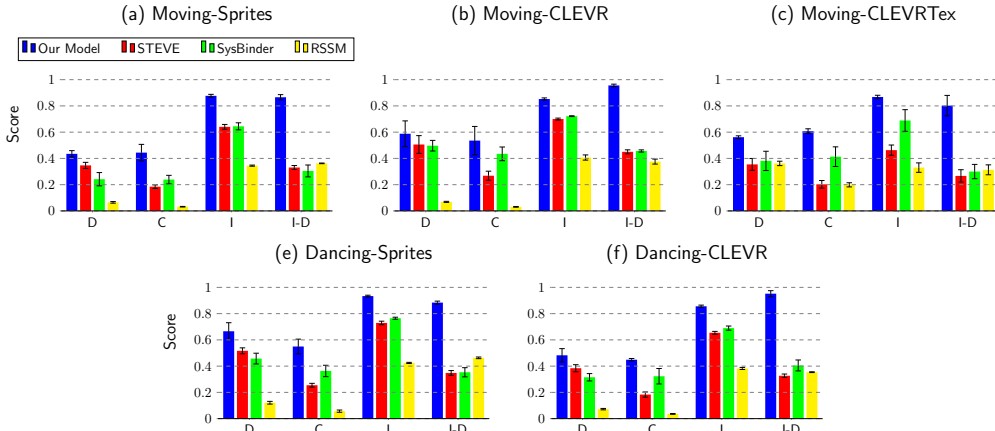

Figure 3: **DCI Performance.** We compare our model with the baselines in terms of Disentanglement (D), Completeness (C), Informativeness (I), and Informativeness-Dynamic (I-D). I-D is the informativeness score for dynamic concepts only (e.g., the direction of motion or dance pattern, etc.) to evaluate how effectively the models capture such dynamic concepts.

## 5.1 Unsupervised Modular Concept Discovery from Videos

**Metrics.** For quantitative evaluation of learned representations, we use the DCI (Eastwood & Williams, 2018) framework to evaluate Disentanglement (D), Completeness (C), and Informativeness (I). Note that the DCI is computed using ground truth object factors that not only include the static factors e.g., color or shape, but also the dynamical factors e.g., direction of motion, "dance" pattern, etc. Additionally, we also compute a metric called *Informativeness-Dynamic* (I-D), which is the informativeness score evaluated only on the dynamic factors.

**DCI Performance.** In Figure 3, we compare the DCI performance of our model with the baselines. We note that our model consistently surpasses the other baselines across all datasets, achieving the highest scores in Disentanglement (D), Completeness (C), and Informativeness (I).

**Emergence of Dynamical Factor Representation.** Importantly, we also note in Figure 3 that our model strongly surpasses all other baselines in terms of the I-D score which captures how informative our representation is about the dynamic factors. Our model achieves scores more than twice those of the other baselines. This strong performance points to the effectiveness of the predictive objective which incentivizes our model to capture dynamical information. As such, to the best of our knowledge, ours is the first unsupervised representation learning model capable of representing both static and dynamic concepts while simultaneously providing disentanglement.

**Visualizing the Feature Space of Blocks.** We visualize the semantics of each block's feature space in Appendix D.1. We can see that our model allocates specific regions of the feature space of specific blocks to capture specific factor values e.g., the star shape, the yellow color, etc. We also see qualitatively that our model captures the dynamical factor values such as the `up&down` dance and the `horizontal sliding` movement.

## 5.2 Compositional Imagination

In this section, we demonstrate how our block-slot representation can be recomposed into novel configurations and be used to generate compositionally novel videos.

**Setup.** We feed the initial frames of a video (called the *context* frames) into our pre-trained RBSU encoder. We take the block-slot representation from the last context frame and manipulate it in one of the following two ways: (1) We can perform a *factor swap* by taking two slots corresponding to distinct objects, selecting the blocks that correspond to a specific factor (e.g., color), and swapping them. (2) We can perform a *factor change* by taking the block-slot representation from a source video, selecting the block that captures a certain factor e.g., a block that represents a certain "dance" movement and using it to replace a block in the block-slot representation of the video that we seek to manipulate. The manipulated block-slot representation is fed to a pre-trained Dreamweaver decoder

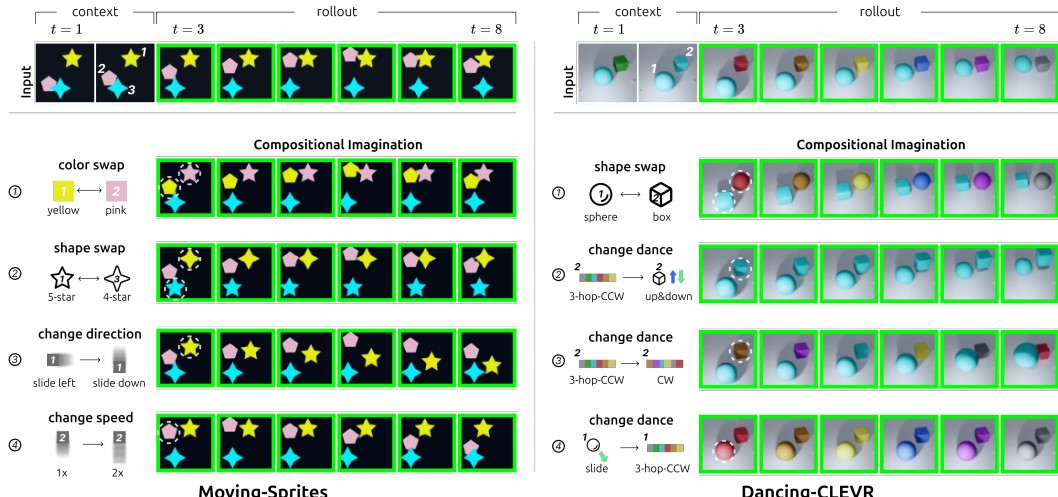

**Figure 4: Compositional Imagination.** We show compositionally novel videos generated by Dreamweaver. In this visualization, we (1) infer the block-slot representation given an initial context video, (2) perform manipulations on the inferred block-slot representation, and (3) perform rollout starting from the manipulated block-slot representation. At the top, we also visualize the rollout that would have occurred had no manipulation been done to the representation. *Left:* For the Moving-Sprites dataset, we visualize manipulations such as swapping color and shape, changing of direction of motion of a specific object, and changing the speed of movement of a specific object. *Right:* For the Dancing-CLEVR dataset, we visualize manipulations such as swapping the object shapes and changing the dance patterns.

to generate future video frames. We autoregressively feed the predicted frames back into the encoder as context frames to perform the rollout. Details about how we ascertain the correspondence between a ground truth factor and its representative block are provided in Appendix C.5.

**Results.** In Figure 4, we can see that our approach successfully generates novel videos through the compositional manipulation of our block-slot representations. For instance, in the Moving-Sprites video, we could create novel object appearances by swapping color or shape blocks (Examples 1 and 2), and alter the objects' motion trajectories and velocities by modifying direction or speed blocks (Examples 3 and 4). Additionally, in the more complex Dancing-CLEVR experiments, we demonstrate the ability to imagine different types of object dynamics, such as transitioning from color-changing dynamics to lifting up and down movement dynamics, highlighting the flexibility of our method (Examples 2 and 4). For more examples of compositional imagination, refer to Appendix D.3. Furthermore, our model effectively generalizes to out-of-distribution block configurations in compositional imagination tasks, preserving quality without degradation. See the detailed results in Appendix D.2.

## 5.3 COMPOSITIONAL SCENE PREDICTION AND REASONING

To evaluate the quality of the learned representations, we construct a downstream task that requires predicting the future states of the objects in the scene and reasoning among the factors of the objects.

**Task.** The task is to take the latent representation of the last frame of a given context video and predict a target value corresponding to a range of future frames at various offsets e.g., $0, \ldots, 5$. The target value corresponding to a frame is defined as follows: (1) We first assign an integer number to each possible value of each ground truth factor following Wu et al. (2024). (2) We then take the maximum value of each ground truth factor across objects and sum these maximum values to obtain the target value. To solve this task well, the learner must, either explicitly or implicitly, predict the future factor values of each object and compare these values to determine the maximum value.

**Probe.** We train a downstream probe that takes as input the latent representation of the last context frame from each pre-trained model and predicts the target value for a range of frame offsets.

**Setup.** We evaluate the probing performance on the Dancing-Sprites and Dancing-CLEVR datasets. For Dancing-Sprites, we use the shape, color, and position of each object as the ground truth factors

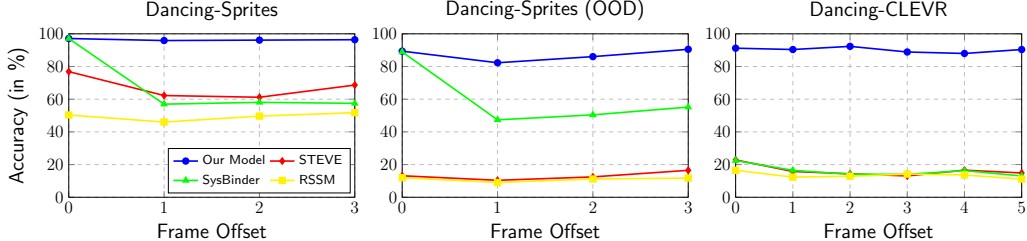

**Figure 5: Compositional Scene Prediction and Reasoning.** We compare our model with baselines in terms of prediction accuracy for different frame offsets. A frame offset of zero corresponds to the last context frame and a frame offset of one corresponds to the first predicted frame after the context frames, and so on.

that determine the target value. Since we provide 3 context frames for this dataset and each object moves in a 4-frame dance sequence, the models must capture the dance patterns to be able to accurately infer the positions. To evaluate out-of-distribution generalization, we further create a Dancing-Sprites (OOD) task where the test set consists only of objects with factor combinations that are not seen during training. For the Dancing-CLEVR dataset, we use the dynamic object movement type as a ground truth factor in addition to the shape, color, and position.

**Probing Performance.** In Figure 5, we can see that Dreamweaver consistently outperforms the baselines on all datasets, indicating the learned block-slot representations are conducive to solving this relational reasoning task.

**Importance of a Predictive Objective.** On Dancing-Sprites, SysBinder can accurately predict the target value at frame offset 0 but the performance degrades for larger frame offsets. This shows the usefulness of Dreamweaver's predictive objective—while SysBinder also decomposes the scene to a block-slot representation corresponding to the factors of the object, it is trained with a reconstruction objective instead of a predictive objective and thus cannot accurately predict the target value for frames outside of the context frames. For Dancing-CLEVR, we see all models besides Dreamweaver fail, even at frame offset zero. This is because the dynamic object movement type is part of the ground truth factors determining the target value, and as we saw from the DCI analysis, the baseline models do not adequately capture these dynamical factors well.

**Generalization Ability.** Dreamweaver and SysBinder generalize well to the Dancing-Sprites-OOD dataset, while the performance of STEVE and RSSM decrease significantly, illustrating the out-of-distribution capability of the block-slot representations. Lastly, we note that the performance of Dreamweaver is generally maintained even as the frame offset is increased beyond the training prediction length. For Dancing-CLEVR, even though we only train the model to predict two frames, we see that the learned representations can be used to predict up to five frames, further indicating that the dynamics of the scene are captured well by the model.

## 5.4 ABLATION STUDY

We conduct a series of ablations on the different architectural components of Dreamweaver as well as several of the key hyperparameters.

**Analysis of Architectural Components.** Figure 6 (a) shows the results of ablating several architectural components on the Dancing-CLEVR dataset. We see that using the concept memory only on the last slot iteration (*CM-on-Last-Slot-It*) performs similarly to our model, which uses the concept memory after every slot iteration, although there is a slight drop in capturing dynamic concepts (I-D). Removing the concept memory completely (*No-CM*), however, results in a sharp drop in disentanglement, showing the importance of using a shared concept memory for our model. Lastly, removing the predictive objective and instead training on reconstruction (*No-Predictive*) results in worse performance across all metrics. In particular, I-D drops significantly, showing the importance of the predictive objective for capturing dynamic concepts.

**Effect of Number of Blocks and Concept Memory Size.** Figures 6 (b-c) show the results of different numbers of blocks and concept memory sizes on the CLEVR-Hard dataset. We see the performance saturating once the number of blocks is large enough, indicating the robustness of this parameter as long as it is set large enough to capture the complexity of the dataset. On the other hand, we observe

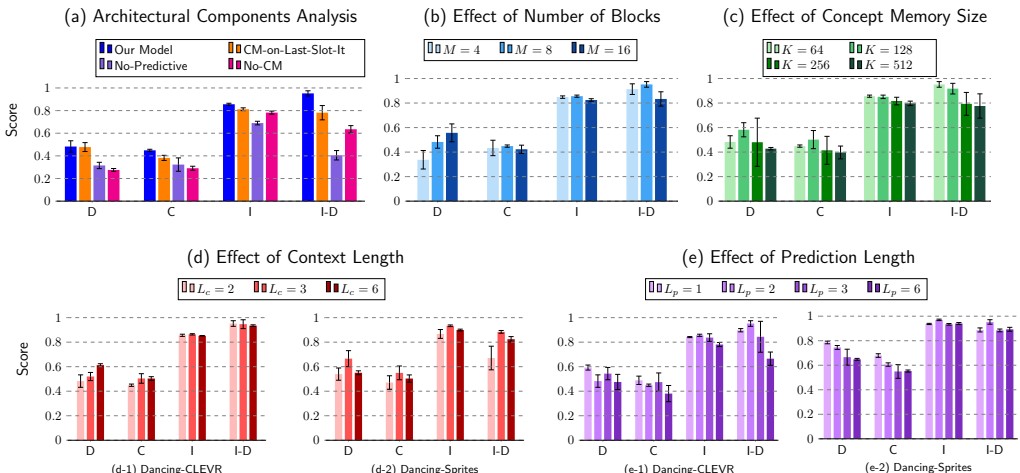

**Figure 6: Ablation Study Results.** *(a)* Architectural ablations for the predictive objective and concept memory on the Dancing-CLEVR dataset. *(b-c)* Varying number of blocks and concept memory size on the CLEVR-Hard dataset. *(d-e)* Varying context length and prediction length on the Dancing-CLEVR and Dancing-Sprites datasets.

that the model's performance is sensitive to the size of the concept memory. Specifically, increasing the concept memory size leads to better disentanglement, but excessively large sizes can degrade performance. This suggests that there is an optimal range for the concept memory size, and finding this balance is crucial for achieving good disentanglement while maintaining overall performance.

**Effect of Context Length and Prediction Length.** Figures 6 (d-e) show the results of varying context and prediction length on the Dancing-CLEVR and Dancing-Sprites datasets. We observe that performance generally improves with longer context lengths, particularly in terms of disentanglement and informativeness. For Dancing-Sprites, there is a significant drop in performance when the context length is less than two, aligning with our design intention that requires observing more than three frames to determine dance patterns accurately, while performance declines again when the context length exceeds four frames. This may be because the dance patterns in this dataset repeat every 4 frames, so a longer context length will not provide more information to the model. We also find that increasing prediction length larger than 2 has a negative affect on performance, particularly for I-D on the Dancing-CLEVR dataset. Since the decoder is shared for the prediction across timesteps, this may be due to the limited capacity of the decoder to represent longer-term predictions.

## 6   LIMITATIONS & CONCLUSION

In this paper, we introduced Dreamweaver, a novel neural architecture for unsupervised learning of compositional world models from videos. The key component of Dreamweaver is the Recurrent Block-Slot Unit, which encodes a temporal context window of past images and maintains a set of independently updated block-slot states, enabling the emergence of abstraction for both static and dynamic concepts. Our model outperforms state-of-the-art object-centric methods and demonstrates the ability to generate videos through novel compositional imagination. This is the first time such compositional imagination has been shown in object-centric learning from videos.

While Dreamweaver represents a significant advancement in the unsupervised learning of compositional world models from visual data, it does have some limitations that future work may address. First, extending Dreamweaver to a probabilistic model could improve its ability to manage uncertainty and generate more diverse and realistic videos. Second, although Dreamweaver has demonstrated the emergence of static and dynamic concepts, exploring the emergence of other abstract concepts, such as numbers, could be fascinating. Lastly, our model shares the general limitation of current state-of-the-art object-centric learning, which is not yet applicable to very complex scene images. Addressing these areas could lead to improved world models.

## REPRODUCIBILITY STATEMENT

We are committed to ensuring the reproducibility of our research. To this end, we intend to make all resources, including the code and datasets specifically designed for this work, publicly available. Prior to release, we will thoroughly verify all implementations and empirical results to guarantee their accuracy and reliability.

## ETHICS STATEMENT

Although we address a fundamental issue in modern deep learning, we are not aware of immediate societal concerns. While carrying deep potential, the practical applications of Dreamweaver remain in developmental stages, indicating that its immediate impact on broader technology or industry sectors is yet to be fully realized. However, future extensions that extend this framework to real-world settings should be mindful of such impact. Also, the environmental impact of training transformer models at scale should be considered in future extensions.

## ACKNOWLEDGEMENTS

This research was supported by the Global Research Development Center (GRDC) through the Cooperative Hub Program (RS-2024-00436165), and by the National Research Foundation of Korea (NRF) through both the Young Researcher Program (No. 2022R1C1C1009443) and the Brain Pool Plus Program (No. 2021H1D3A2A03103645), funded by the Ministry of Science and ICT.

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

## A    ADDITIONAL RELATED WORKS

**Learning Compositional Mechanisms.** Unlike our model is built on slot-based models, there is another line of work for learning compositional representations motivated by the *Independent Causal Mechanisms* principle and the *Sparse Mechanism Shift* hypothesis (Schölkopf et al., 2012; 2021b). RIMs architecture family (Goyal et al., 2021b; 2020; Madan et al., 2021; Assouel et al., 2022) decomposes state space representations into separate recurrent components that operate independently, with sparse interactions among them. These models are designed to learn a set of reusable mechanisms that are selectively activated through sparse communication, enabling them to capture recurring concepts across frames. Similarly, our model also learns reusable and modularized representations, however, our representations are always used and more structured by introducing a hierarchical structure, where these blocks are encapsulated within individual slot representations. Lastly, NPS (Goyal et al., 2021a) model, as a recent work of RIMs, learns a set of independent mechanisms capturing the interaction between objects, while ours is still limited in this ability.

**Compositional World Models.** Previous work has studied world models from the perspective compositional generalization (Zhao et al., 2022; Sehgal et al., 2024; Zhou et al., 2024). In particular, Cosmos (Sehgal et al., 2024) is a framework for object-centric world modeling that leverages a frozen pretrained vision-language foundation model to decompose objects into symbolic attributes. Du et al. (2023); Zhou et al. (2024); Yu et al. (2022); Cho et al. (2024) similarly relies on language to enable compositional generalization. Our model, on the other hand, is able to learn a compositional world model from pixels without the use language input or pretrained foundation models. COMET (Lei et al., 2024) is also a related model that learns disentangled modes of interaction between objects, but does not learn attribute-level representations of the objects.

## B    ADDITIONAL MODEL DETAILS

### B.1    IMAGE TOKENIZATION VIA DISCRETE VAE

Since we leverage an autoregressive image transformer to predict the frames $\mathbf{x}_{t+1}, \ldots, \mathbf{x}_{t+K}$, we train a discrete Variational Autoencoder (dVAE) to generate discrete token representations of these frames and act as prediction targets. The dVAE consists of an encoder $f_\phi^{\text{dVAE}}$ and a decoder $g_\theta^{\text{dVAE}}$. The dVAE encoder takes an image as input and outputs a sequence of patch-level tokens or latent codes $z_{t,1}, \ldots, z_{t,L'}$ where $z_{t,l} \in \mathcal{I}$, while the dVAE decoder takes a sequence of tokens as input and decodes it to an image. The dVAE encoder and decoder are trained with an autoencoding objective as follows:

$$z_{t,1}, \ldots, z_{t,L'} = f_\phi^{\text{dVAE}}(\mathbf{x}_t) \implies \hat{\mathbf{x}}_t = g_\theta^{\text{dVAE}}(z_{t,1}, \ldots, z_{t,L'}) \implies \mathcal{L}_{\text{dVAE}}(\theta, \phi) = ||\hat{\mathbf{x}}_t - \mathbf{x}_t||_2^2,$$

where we use Gumbel-Softmax relaxation for the discrete latent codes to facilitate training and a simple squared error as the reconstruction loss.

### B.2    CONDITIONAL PREDICTION WITH STEP INDICATORS

We train our model to predict multiple future frames $\mathbf{x}_{t+1}, \ldots, \mathbf{x}_{t+K}$, which encourages the encoder $f_\phi$ to extract consistent dynamic features by preventing the model from overfocusing on image generation. However, predicting all frames simultaneously is inefficient. Instead, we improve efficiency by randomly sampling one frame $\mathbf{x}_{t+k}$ to predict during each training step. This is achieved through conditional prediction using a single transformer decoder $g_\theta$ and a step indicator $i_k$.

**BOS token as a step indicator.** In Dreamweaver, the Beginning-of-Sequence (BOS) token in the transformer decoder functions as a step indicator $k$. Specifically, we initialize a distinct $\text{BOS}_{\theta,k}$ token for each step $k$ and input the corresponding $\text{BOS}_{\theta,k}$ token, based on the index $k$ sampled during each training step, as the first token for autoregressive generation.

**Self-modulation with step indicators.** In more complex and realistic datasets, relying solely on the BOS token limits conditional prediction performance. To address this, we employ the Self-modulation technique (Karras et al., 2019; Lee et al., 2021). Specifically, we enhance the transformer decoder $g_\theta$ by replacing every layer normalization with self-modulated layer normalization (SLN), computed as

follows:

$$\text{SLN}(h_\ell, \mathbf{w}_{\theta,k}) = \gamma_\ell(\mathbf{w}_{\theta,k}) \odot \frac{\mathbf{h}_\ell - \mu}{\sigma} + \beta_\ell(\mathbf{w}_{\theta,k})$$

where $h_\ell$ is the input from the previous layer, $\mu$ and $\sigma$ denote the mean and variance of the inputs within the layer, while $\gamma_\ell$ and $\beta_\ell$ compute the adaptive normalization parameters. $\mathbf{w}_{\theta,k}$ is a learnable latent vector used for guiding explicit modularized generation.

## C  ADDITIONAL IMPLEMENTATION DETAILS

### C.1  TRAINING AND IMPLEMENTATION DETAILS

We utilized images with a 64x64 resolution for all datasets except Moving-CLEVRTex, which uses a 128x128 resolution. Each model was trained on NVIDIA GeForce RTX 4090 GPUs with 24GB of memory. The Dreamweaver model underwent 400,000 iterations of training, taking 20 hours for the Moving-Sprites and Moving-Sprites-OOD datasets, 30 hours for the Dancing-Sprites, Moving-CLEVR, and Dancing-CLEVR datasets, and 48 hours for Moving-CLEVRTex. Additionally, we employed the self-modulation technique for conditional prediction exclusively in the Moving-CLEVRTex dataset.

### C.2  HYPERPARAMETERS

Table 1 details the hyperparameters employed for the various datasets in our Dreamweaver experiments. We trained all models using the Adam optimizer (Kingma & Ba, 2014) with $\beta_1$ set to 0.9 and $\beta_2$ set to 0.999. Furthermore, we utilized the architecture and hyperparameters of the backbone image encoder as specified in Singh et al. (2023).

**Table 1:** Hyperparameters of our model used in our experiments. We use a shortened version of the dataset name, omitting prefixes such as "Moving-" or "Dancing-" unless they are necessary to distinguish between datasets.

| | | Dataset | | | |
|---|---|---|---|---|---|
| Module | Hyperparameter | Moving-Sprites | Dancing-Sprites | CLEVR | CLEVRTex |
| General | Batch Size | 24 | 24 | 24 | 48 |
| | Training Steps | 400K | 400K | 400K | 400K |
| | Image Size | $64 \times 64$ | $64 \times 64$ | $64 \times 64$ | $128 \times 128$ |
| | Context Length, $T$ | 2 | 3 | 2 | 2 |
| | Prediction Length, $K$ | 2 | 3 | 2 | 2 |
| | Grad Clip (norm) | 0.5 | 0.5 | 0.5 | 0.5 |
| RBSU | # Iterations | 3 | 3 | 3 | 3 |
| | # Slots | 5 | 5 | 5 | 5 |
| | # Prototypes | 64 | 64 | 64 | 128 |
| | # Blocks | 8 | 8 | 8 | 8 |
| | Block Size | 96 | 96 | 96 | 96 |
| | Learning Rate | 0.00005 | 0.00005 | 0.00005 | 0.00005 |
| Discrete VAE | Patch Size | $4 \times 4$ | $4 \times 4$ | $4 \times 4$ | $4 \times 4$ |
| | Vocabulary Size | 4096 | 4096 | 4096 | 4096 |
| | Temp. Start | 1.0 | 1.0 | 1.0 | 1.0 |
| | Temp. End | 0.1 | 1.0 | 0.1 | 0.1 |
| | Temp. Decay Steps | 60K | 60K | 60K | 60K |
| | Learning Rate | 0.0003 | 0.0003 | 0.0003 | 0.0003 |
| Transformer Decoder | # Layers | 8 | 4 | 8 | 8 |
| | # Heads | 4 | 4 | 4 | 4 |
| | Hidden Size | 192 | 192 | 192 | 192 |
| | Dropout | 0.1 | 0.1 | 0.1 | 0.1 |
| | Learning Rate | 0.0003 | 0.0003 | 0.0003 | 0.0005 |

## C.3 BASELINES

In this paper, we train three baseline models: STEVE, SysBinder, and RSSM. For STEVE, we use the original architecture and hyperparameters, modifying only the slot size to 64 dimensions. For SysBinder, we adapt the architecture to handle video data with a recurrent structure and align the hyperparameters with those of our model for a more accurate comparison, including block size, the number of prototypes, and the number of blocks. For RSSM, we implement the world model from DreamerV1 (Hafner et al., 2019), using 32 dimensions for both deterministic and stochastic latents, while adhering to the other recommended hyperparameters.

For implementation, we utilize the following open-source resources:

- STEVE (Singh et al., 2022b): https://github.com/singhgautam/steve
- SysBinder (Singh et al., 2023): https://github.com/singhgautam/sysbinder
- RSSM (Hafner et al., 2020): https://github.com/jurgisp/pydreamer

## C.4 DOWNSTREAM MODEL ARCHITECTURE

For the downstream experiments on all models except for RSSM, we use a transformer architecture with 4 layers, 4 heads, 0.1 dropout, and model dimension of 128. We use the output of a learned class token to predict the target label.

For RSSM, we use an MLP to make the prediction. While we experimented with matching the number of parameters with the transformer model, we found better performance using a smaller network with 4 layers and hidden dimension of 128 and ReLU activation.

During training, we freeze the up-stream models and train with a learning rate of 3e-4.

## C.5 IDENTIFYING BLOCK-FACTOR CORRESPONDENCE.

To swap or change a factor, it is important to identify which block index corresponds to which object factor. For this, we adopt the following procedure. (1) Take a large batch of videos with per-object factor labels given. (2) Extract block-slot representation from the videos using a pre-trained RBSU encoder. (3) Match slots with ground truth labels via Hungarian matching using mask overlap. (4) Train a probe to predict ground truth factor labels from the block-slot representation. Use probing methods that provide feature importance (e.g., LASSO or Decision Trees). (5) Manually inspect the feature importances to identify which block(s) represent a specific ground truth factor.

# D ADDITIONAL EXPERIMENT RESULTS

## D.1 VISUALIZATION OF CAPTURED CONCEPTS

Figure 7 and 8 illustrates the concepts represented by each block within learned block-slot representations. To achieve this, we gather block representations with the same index and apply clustering methods, such as $k$-means, following the approach in Singh et al. (2023).

For instance, in the Moving-Sprites dataset, `block 0` captures shape concepts such as `Star` and `Square`, `block 2` captures color concepts such as `Yellow` and `Red`, and `block 1` captures dynamic concepts such as `Lift Up & Down` and `Slide Leftside & Rightside` motion. Similarly, in the Dancing-CLEVR dataset, `block 7` captures shape concepts, `block 0` captures color concepts, and `block 4` captures dynamic concepts.

## D.2 OUT-OF-DISTRIBUTION COMPOSITIONAL IMAGINATION

In Section 5.2, we demonstrate that our model is capable of synthesizing novel videos by recombining block-slot representations. In this section, we further investigate whether Dreamweaver can generate videos featuring objects with compositionally out-of-distribution (OOD) properties. To this end, we introduce the Moving-Sprites-OOD dataset, a modified version of the Moving-Sprites dataset. In this version, the model is trained with only 90% of the possible (shape, color, moving direction)

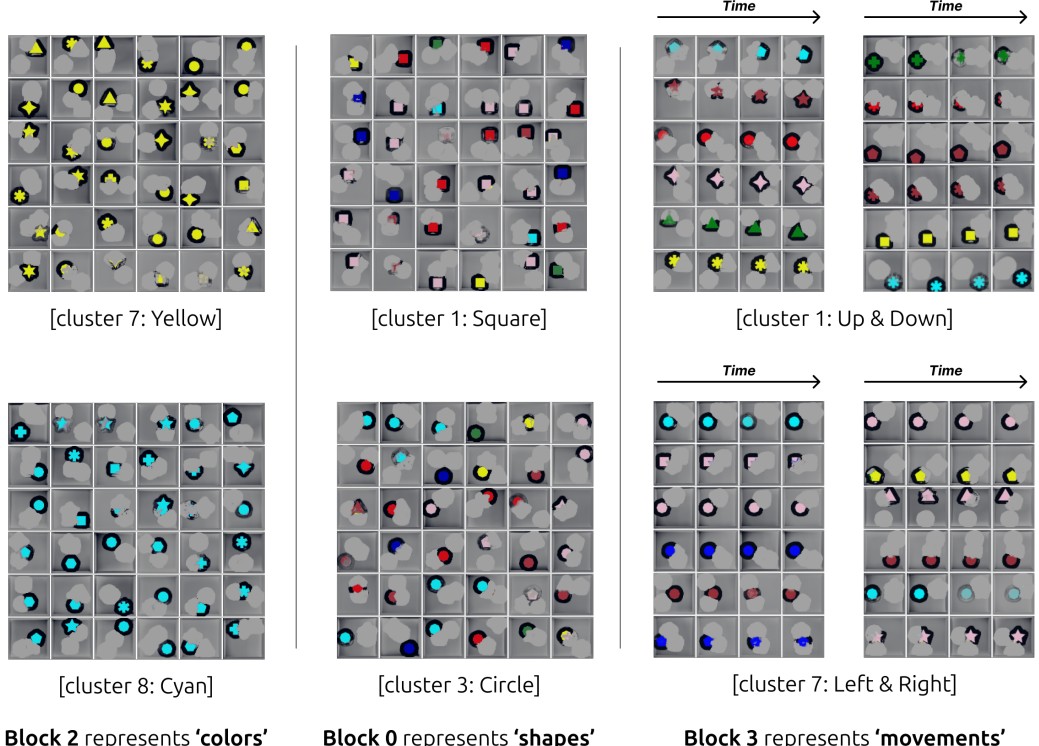

Figure 7: Visualization of Captured Concept in Moving-Sprites dataset.

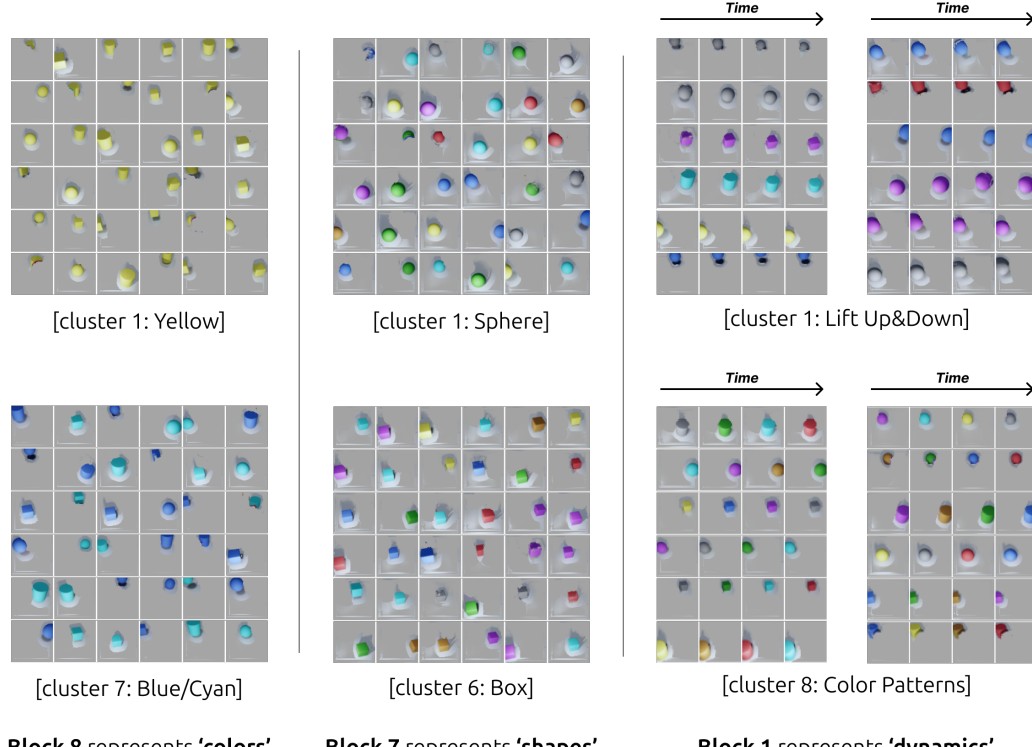

Figure 8: Visualization of Captured Concept in Dancing-CLEVR dataset.

combinations, deliberately withholding the remaining 10% to test the model's ability to generalize to unseen combinations.

To evaluate out-of-distribution (OOD) compositional imagination, we adhere to the methodology outlined in Section 5.2. As a result, our model successfully generates novel video frames that feature objects with properties not present in the training dataset. For instance, as shown in Figure 9, we intentionally modify a block-slot representation of an object characterized by (brown, square, move left) — an object seen during training — into (cyan, square, move left) by replacing the brown block with a cyan block, which is an unseen combination. Consequently, we observed that our model extends its compositional imagination capabilities to novel, compositionally unseen objects. Additional examples are provided in Figure 9.

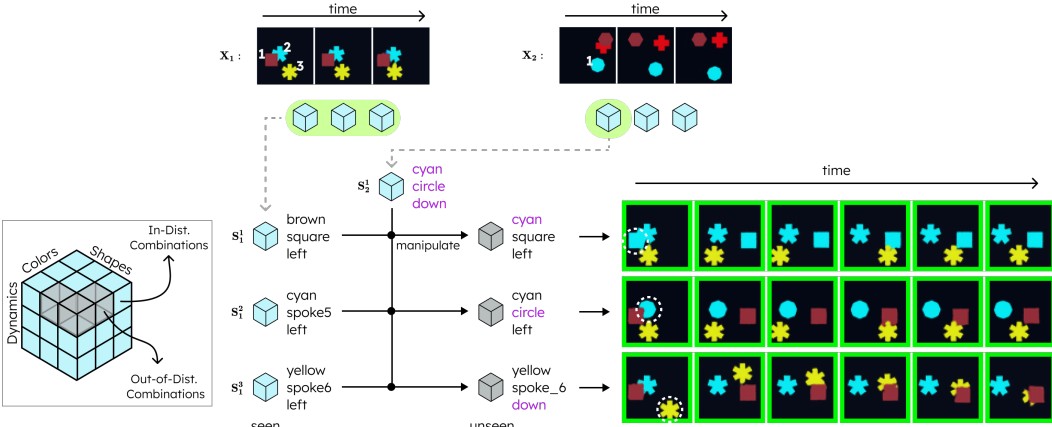

**Figure 9: Out-of-Distribution Compositional Imagination Example.**

## D.3 ADDITIONAL EXAMPLES OF COMPOSITIONAL IMAGINATION TASKS

To further illustrate the concept of compositional imagination, we provide additional examples of tasks that demonstrate this capability. Some examples from Moving-CLEVRTex dataset are visualized in Figure 10.

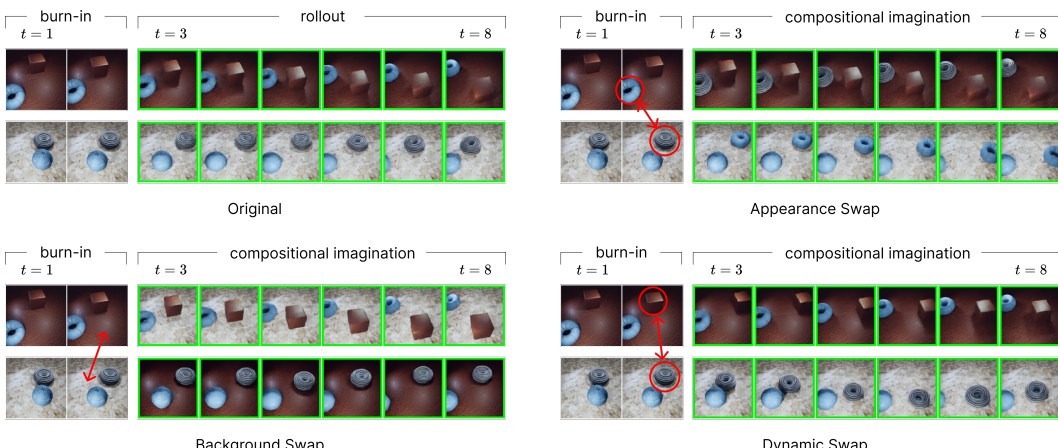

**Figure 10: Compositional imagination examples on Moving-CLEVRTex.** In this visualization, we demonstrate the generation of compositionally novel videos on visually more complex and textured scenes than the previously tested datasets. *Top-Left:* We show two original videos from the dataset. *Top-Right:* We swap the block representations of object texture between two objects across videos after the context phase and roll-out. *Bottom-Left:* We swap the background blocks of the two videos after the context phase and roll-out. *Bottom-Right:* We swap the dynamics-capturing blocks of two objects across the two videos after the context phase and roll-out.

### D.4 Additional Quantitative Results for Future Imagination

In addition to the remarkable qualitative outcomes of compositional imagination, we present quantitative results in Table 2 to provide a more comprehensive understanding of our work. We computed MSE, LPIPS (Zhang et al., 2018), and PSNR (Hore & Ziou, 2010) to evaluate compositional imagination, where the blocks are manipulated in the final context frame and the remaining video is rolled out. Specifically, for each test video, we performed the following manipulation: we swapped the blocks for each intra-object factor across all object pairs in the video. Alongside, for each of these manipulations, we generated ground truth videos also. To assess the quality of the imagined compositions, we calculated MSE, LPIPS, and PSNR metrics by comparing the compositionally imagined videos to their respective ground truth videos.

**Table 2: Quantitative Results for Compositional Imagination Performance.** We calculate this metric by averaging over 20 different video scenarios, automatically conducting all semantically possible combinations of block-slot manipulations per each video scenario.

|  | Dataset | |
|---|---|---|
|  | **Moving-Sprites** | **Dancing-CLEVR** |
| **MSE** | 1261 | 1484 |
| **LPIPS** | 0.288 | 0.395 |
| **PSNR** | 30.42 | 33.83 |

### D.5 Generalization to Entirely Unseen Concepts

In order to further evaluate the out-of-distribution generalization ability of Dreamweaver, we ran additional experiments on the Dancing Sprites dataset for the downstream task from Section 5.3 with novel shapes and dance patterns that were not seen when training the underlying baseline and Dreamweaver models. In this setup, we use the same pre-trained models as used in Section 5.3, but introduce 6 novel shapes (OOD Shapes) and 4 novel dance patterns (OOD Dynamics) when training and evaluating the downstream probe. For OOD Shapes, we introduce the following shapes: `hexagon`, `octagon`, `star_5`, `star_6`, `spoke_4`, `spoke_6`. For OOD Dynamics, we show the novel dance patterns in Figure 11. We show the results of these experiments in Figure 12. We see that Dreamweaver still outperforms the baselines in this setting and we do not see much performance degradation when compared to the in distribution setting (Figure 5 (left)). This indicates that the block-slot representations can be useful for downstream tasks even on data with static and dynamic concepts not previously seen during pre-training.

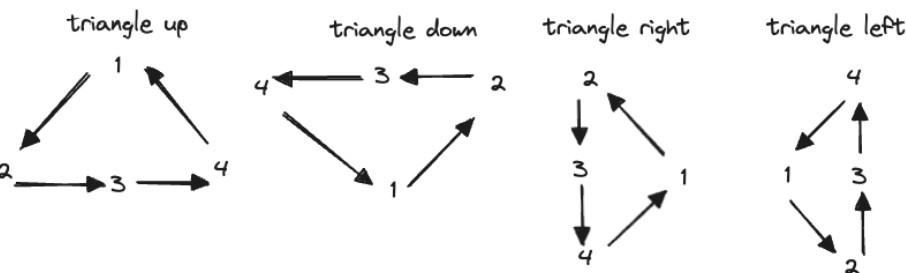

**Figure 11: Visualization of Unseen Dance Patterns in Dancing-Sprites (OOD Dynamics) Experiments**

## E Details of Dataset Design

We provide detailed information that is carefully considered in our dataset design. For a comprehensive understanding of our design, refer to the visualized overview in Appendix 13.

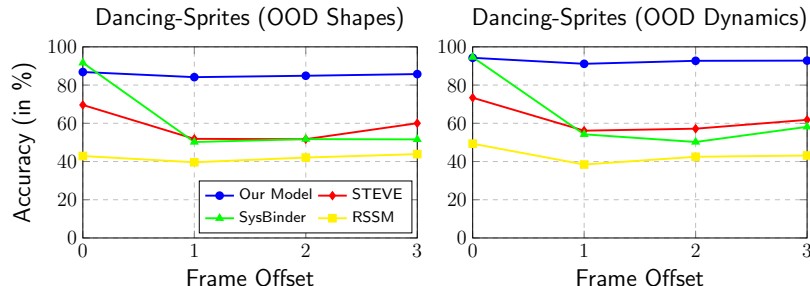

Figure 12: Downstream Performance with Entirely Unseen Concepts

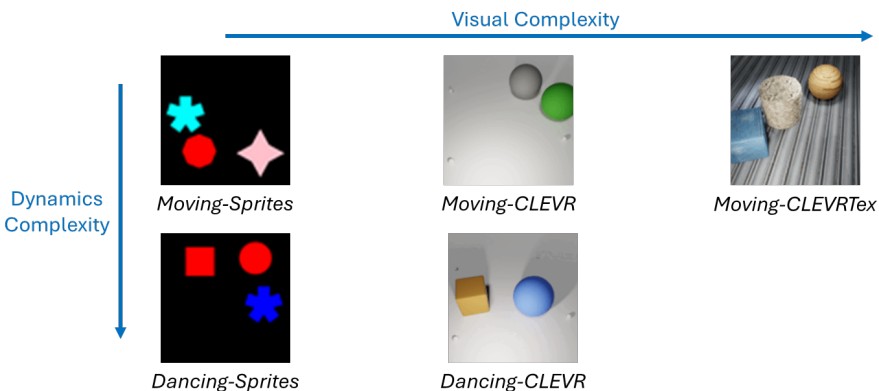

Figure 13: Overview of Designed Datasets

### E.1    SIMPLE DYNAMIC DATASETS

For the datasets with simple dynamics, we incorporate sliding movements into the existing static datasets. Moving-Sprites features three fixed-sized objects with 12 shapes and 7 colors sliding in 4 directions at 4 different speeds within a 2D scene (detailed in Table 3). Moving-CLEVR consists of two or three fixed-sized objects with 3 shapes and 8 colors sliding in 4 directions at a constant speed within a 3D scene (detailed in Table 4). Moving-CLEVRTex includes two or three objects of 2 different sizes, each with one of 4 shapes and 10 complex textures, moving on a textured ground with the same dynamics as Moving-CLEVR (detailed in Table 5). Within an episode, multiple objects simultaneously evolve under their corresponding dynamics.

### E.2    ADVANCED DYNAMIC DATASETS

For the datasets with advanced dynamics, we introduce additional complexity beyond sliding movements. Dancing-Sprites includes three fixed-size objects derived from 6 shapes and 10 colors, each performing one of 4 patterned dances (detailed in Table 6). Each dance pattern comprises a sequence of 4 movement steps, requiring models to understand dynamics over longer context frames. Dancing-CLEVR builds upon Moving-CLEVR by adding two extra motions: 'lift motion', where objects move vertically instead of sliding, and 'color pattern dynamics', where each object follows one of 4 distinct color-changing patterns over time (detailed in Table 7). Similar to simple dynamic dataset, multiple objects simultaneously evolve under their corresponding dynamics within an episode.

**Table 3: Primitive Object Factors in Moving-Sprites.** In addition to their combinations of primitive properties, all objects in the Moving-Sprites datasets have a fixed size of 0.22.

| Shape | Color (RGB) | Moving Direction | Speed |
|---|---|---|---|
| square | (0, 0, 255) | Up | 0.025 |
| triangle | (0, 128, 0) | Down | 0.05 |
| star_4 | (255, 255, 0) | Left | 0.075 |
| star_5 | (255, 0, 0) | Right | 0.1 |
| star_6 | (0, 255, 255) | | |
| circle | (255, 192, 203) | | |
| pentagon | (165, 42, 42) | | |
| hexagon | | | |
| octagon | | | |
| spoke_4 | | | |
| spoke_5 | | | |
| spoke_6 | | | |

**Table 4: Primitive Object Factors in Moving-CLEVR.** In addition to their combinations of primitive properties, all objects in the Moving-CLEVR datasets have a fixed size of 1.6, are made of rubber material, and move at a constant speed.

| Shape | Color (RGB) | Moving Direction |
|---|---|---|
| Cube | (87, 87, 87) | Forward |
| Sphere | (173, 35, 35) | Backward |
| Cylinder | (42,75, 215) | Leftside |
| | (29, 105, 20) | Rightside |
| | (129, 74, 25) | |
| | (129, 38, 192) | |
| | (41, 208, 208) | |
| | (255, 238, 51) | |

**Table 5: Primitive Object Factors in Moving-CLEVRTex.** All objects in Moving-CLEVRTex slide at a fixed speed. For materials, we use free textures provided by Polyhaven. You can directly download through clevrtex_generation (Karazija et al., 2021) code (link).

| Shape | Materials | Size | Moving Direction |
|---|---|---|---|
| Cube | whitemarble | 1.6 | Forward |
| Sphere | polyhaven_leather_red_02 | 2.0 | Backward |
| Cylinder | polyhaven_factory_wall | | Leftside |
| Torus | polyhaven_cracked_concrete_wall | | Rightside |
| | poly_haven_stony_dirt_path | | |
| | polyhaven_painted_metal_shutter | | |
| | polyhaven_raw_plank_wall | | |
| | polyhaven_denim_fabric | | |
| | polyhaven_large_grey_tiles | | |
| | polyhaven_medieval_blocks_02 | | |

**Table 6: Primitive Object Factors in Dancing-Sprites.** Along with their combinations of primitive properties, all objects in the Dancing-Sprites datasets maintain a fixed size of 0.22 and move at a constant speed of 0.15. For more details on dance patterns, see Figure 14.

| Shape | Color (RGB) | Pattern |
|---|---|---|
| square | (0, 0, 255) | Clockwise_Square |
| triangle | (0, 128, 0) | Counter_Clockwise_Square |
| star_4 | (255, 255, 0) | Clockwise_Diamond |
| circle | (255, 0, 0) | Counter_Clockwise_Diamond |
| pentagon | (0, 255, 255) | |
| spoke_5 | (255, 192, 203) | |
| | (165, 42, 42) | |
| | (0, 255, 0) | |
| | (255, 0, 255) | |
| | (255, 165, 0) | |

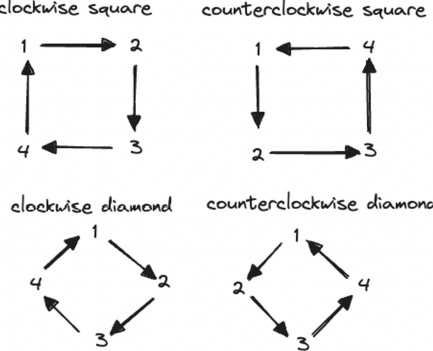

**Figure 14: Visualization of Dance Patterns in Dancing-Sprites.** There are four distinct dance patterns, each composed of a sequence of four movements. Specifically, Clockwise_Square follows [right, down, left, up]; Counter_Clockwise_Square follows [left, down, right, up]; Clockwise_Diamond follows [downright, downleft, upleft, upright]; Counter_Clockwise_Diamond follows [downleft, downright, upright, upleft].

**Table 7: Primitive Object Factors in Dancing-CLEVR.** This dataset retains all object properties from Moving-CLEVR, with the addition of new dynamic patterns. The maximum height of the lifting motion is twice the object's height. For more details on color patterns, see Figure 15.

| Shape | Color (RGB) | Dynamics |
|---|---|---|
| Cube | (87, 87, 87) | Slide Forward |
| Sphere | (173, 35, 35) | Slide Backward |
| Cylinder | (42, 75, 215) | Slide Leftside |
| | (29, 105, 20) | Slide Rightside |
| | (129, 74, 25) | Color Pattern: CW |
| | (129, 38, 192) | Color Pattern: CCW |
| | (41, 208, 208) | Color Pattern: 3-hop CW |
| | (255, 238, 51) | Color Pattern: 3-hop CCW |
| | | Lift Up |
| | | Lift Down |

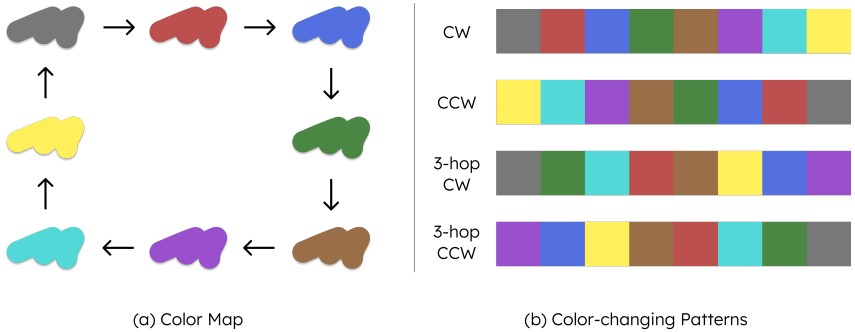

(a) Color Map            (b) Color-changing Patterns

**Figure 15: Visualization of Color-changing Patterns in Dancing-CLEVR.** The dataset includes four distinct color-changing patterns, as illustrated in (b). Each sequence is generated by following the color map in (a): (1) Clockwise Selection (CW), (2) Counter-Clockwise Selection (CCW), (3) 3-hop Clockwise Selection (3-hop CW), and (4) 3-hop Counter-Clockwise Selection (3-hop CCW).

