# OpenReview forum: "Dreamweaver: Learning Compositional World Models from Pixels"
_ICLR.cc/2025/Conference — ICLR 2025 Poster_

### Official Review · Reviewer_ZQtq · 2024-11-02

**Soundness:** 3
**Presentation:** 4
**Contribution:** 2
**Rating:** 6
**Confidence:** 4

**Summary:**

This paper studies the problem of discovering shared visual concepts from videos and recomposing the concepts to unseen videos. The authors propose a novel neural architecture, Dreamweaver, operating on video object-centric representations. In Dreamweaver, a novel Recurrent Block-Slot Unit (RBSU) decomposes videos into objects and attributes and a multi-future-frame prediction loss captures disentangled representations to form both dynamic and static concepts. Experiments demonstrate that Dreamweaver can outperform current SOTA baselines on DCI scores of multiple datasets. The visualization experiments are conducted to show the compositional imagination ability of Dreamweaver as well.

**Strengths:**

The paper is well-written. The authors propose a method that learns a list of static and dynamic abstract concepts from videos. The learned concepts are interpretable, and Dreamweaver can imagine unseen videos by applying the learned concepts to new objects. The authors conduct extensive experiments to show the effectiveness of the proposed method compared with baselines.

**Weaknesses:**

My major concern is the generalizability of the proposed method. Several design choices prevent Dreamweaver from discovering latent rules beyond the datasets used in the paper. Dreamweaver assumes the videos only involve single-object moving, which significantly constrains the possible dynamic Dreamweaver can represent. It is unclear how the architecture can be modified to relax this assumption. The model also assumes a predefined set of prototypes that will be seen during training. The only generalization we would see in the test split is a novel combination of prototypes and objects, which is not surprising in slot-based architectures trained on synthetic datasets.

**Questions:**

1.	Did the authors try increasing the number of prototypes? I notice that currently, the datasets use different numbers of prototypes. Is the number of prototypes an important hyperparameter for Dreamweaver? The assumption of knowing the number of prototypes in the dataset makes the setting less realistic. Hence it would be important to show whether the method can have reasonable performance when the predefined number is not equal to the total number of rules in the dataset.
2.	Could Dreamweaver be extended to discover and simulate multi-object physical concepts like object collision? It seems that the objects cannot interact with each other in current settings.

---

> ### Author Response · Authors · 2024-11-25
>
> *We would like to thank the reviewer for their thoughtful review and a positive recommendation!*
>
> ---
>
> ### R1. Regarding Single Moving Object
>
> > My major concern is the generalizability of the proposed method. Several design choices prevent Dreamweaver from discovering latent rules beyond the datasets used in the paper. Dreamweaver assumes the videos only involve single-object moving, which significantly constrains the possible dynamic Dreamweaver can represent. It is unclear how the architecture can be modified to relax this assumption.
> >
>
> It is not true that our videos have only a single moving object. We believe there has been a misunderstanding regarding the capabilities of our model that we would like to address.
>
> Our datasets and model actually support multiple simultaneously moving and changing objects. All our experimental scenes contain 2-3 objects that move and transform concurrently, and we have now enhanced the dataset description in Appendix E to make this clearer.
>
> Also, to our knowledge, there is nothing in our approach that prevents it from scaling to any arbitrary number of dynamic objects. In fact, one defining characteristic of our model is that it can learn disentangled representations from multiple objects and multiple object attributes. Please let us know which Dreamweaver’s design choices made you think otherwise—we are happy to address this further.
>
> ---
>
> ### R2. Regarding Predefined Prototypes
>
> > The model also assumes a predefined set of prototypes that will be seen during training. The only generalization we would see in the test split is a novel combination of prototypes and objects, which is not surprising in slot-based architectures trained on synthetic datasets.
> >
>
> We sincerely appreciate your careful review of our work.
> However, it is not true that our model uses a predefined set of prototypes. we believe there are some misconceptions about our architecture.  We would like to kindly clarify an important aspect of our model's design: our model employs learnable vectors as prototypes in its concept memory, which learn to represent semantic concepts through backpropagation, rather than relying on any predefined set of prototypes. We will ensure this crucial distinction is more clearly articulated in our revised paper. (See Section 2.2)
>
> Regarding the generalization capabilities of our model, we would like to elaborate on several important points:
> First, in this paper, we focus on systematic generalization - specifically, the ability to generalize to novel combinations of familiar concepts. While this might seem straightforward, it remains one of the fundamental challenges in deep learning, as demonstrated by recent literature [1, 2, 3]. We believe this challenge merits continued attention and investigation.
>
> Second, while we acknowledge and appreciate the successful demonstrations of systematic generalization by slot-based methods [4, 5], our work makes a distinct contribution by extending beyond visual factors (such as color or shape) to address dynamical factors (such as dance patterns). To our knowledge, this represents the first demonstration of systematic generalization in the domain of dynamical factors, which we consider a meaningful advancement in the field.
>
> Finally, we are encouraged to share additional experimental results from Appendix Section D.5, where we evaluated our model on downstream scene prediction and reasoning tasks using shapes and dance patterns that were not present during pre-training. In these out-of-distribution (OOD) scenarios, our model demonstrated superior performance compared to all baselines. These results suggest that Dreamweaver's block-slot representations effectively generalize to downstream tasks, even when encountering previously unseen static and dynamic concepts.
>
> We appreciate the opportunity to clarify these points and will ensure they are more clearly presented in our revised manuscript.
>
> [1] Berglund, Lukas, et al. "The reversal curse: Llms trained on" a is b" fail to learn" b is a"."
>
> [2] Lake, Brenden, and Marco Baroni. "Generalization without systematicity: On the compositional skills of sequence-to-sequence recurrent networks."
>
> [3] Kim, Yeongbin, et al. "Imagine the unseen world: a benchmark for systematic generalization in visual world models."
>
> [4] Singh, Gautam, Fei Deng, and Sungjin Ahn. "Illiterate dall-e learns to compose."
>
> [5] Singh, Gautam, Yeongbin Kim, and Sungjin Ahn. "Neural systematic binder."

---

> > ### Author Response · Authors · 2024-11-25
> >
> > ### R3. Assumption of Knowing the Number of Prototypes
> >
> > > Did the authors try increasing the number of prototypes? I notice that currently, the datasets use different numbers of prototypes. Is the number of prototypes an important hyperparameter for Dreamweaver? The assumption of knowing the number of prototypes in the dataset makes the setting less realistic. Hence it would be important to show whether the method can have reasonable performance when the predefined number is not equal to the total number of rules in the dataset.
> > >
> >
> > We would respectfully like to point out that this seems to be a **misunderstanding**. We do not make any assumptions about the true number of factor values in the datasets.
> >
> > Instead, we consistently use a significantly larger number of prototypes (typically 64 or 128) compared to the actual number of factor values present in our datasets (which usually range from 3 to 10). This design choice ensures our method remains practical and applicable to real-world scenarios where the true number of factors may be unknown.
> >
> > For a more detailed analysis of how the number of prototypes affects our model's performance, we invite you to review Figure 6 and Section 5.4 of our paper, where we provide a comprehensive examination of this relationship. Note that in the figure, we use the term *concept memory size* instead of *number of prototypes*. We apologize for any confusion this may have caused and will ensure this terminology is clarified in the final version of the paper.
> >
> > ---
> >
> > ### R4. Extension to Datasets with Object Interactions
> >
> > > Could Dreamweaver be extended to discover and simulate multi-object physical concepts like object collision? It seems that the objects cannot interact with each other in current settings.
> > >
> >
> > Thank you for this insightful question about extending Dreamweaver to handle multi-object physical interactions. We are excited about the potential directions you've highlighted!
> >
> > In our current work, we took a foundational first step by focusing on discovering dynamical factors in scenes without object interactions, as this represents the first attempt in this research direction. Looking forward, we see promising opportunities to expand Dreamweaver's capabilities to more complex scenarios involving object-object interactions. This could include discovering and modeling physical properties such as mass, elasticity, and friction that govern how objects interact with each other.

---

> > > ### Comment · Area_Chair_KYjh · 2024-11-30
> > >
> > > Dear Reviewer,
> > >
> > > The authors have provided their responses. Could you please review them and share your feedback?
> > >
> > > Thank you!

---

> ### Comment · Reviewer_ZQtq · 2024-12-01
>
> I thank the authors for the detailed clarification. The response has resolved my main concerns. Based on the provided results, I would like to keep my score at 6.

---

### Official Review · Reviewer_aS2J · 2024-11-02

**Soundness:** 3
**Presentation:** 2
**Contribution:** 2
**Rating:** 6
**Confidence:** 3

**Summary:**

1. The authors propose a novel method Dreamweaver for learning composable concepts(static and dynamic) from videos in an unsupervised way.
2. The authors introduce motion into existing 2d and 3d static datasets at two different complexities (simple and advanced)
3. The authors demonstrate the effectiveness of their method on their datasets along three axes of comparison; concept discovery, compositional generation and out-of-distribution generalisation.

**Strengths:**

1. The authors are the first to introduce a method to learning dynamic composable concepts from videos in an unsupervised way on top of static composable concepts while maintaining disentanglement.
2. The authors introduce a novel module Recurrent Block Slot Unit to model dynamic concepts.
3. Instead of the traditional reconstruction objective, the authors use a predictive objective to model dynamic concepts better.
4. The authors demonstrate the effectiveness of their method on their datasets along three axes of comparison; concept discovery, compositional generation and out-of-distribution generalisation.

**Weaknesses:**

1. The compositional imagination evaluation only has qualitative results which while interesting is not very informative about the model's performance relative to the other baselines. Some comparative, quantitative results should help here. For example, the authors can holdout a set of combinations in their dataset during training and evaluate the fidelity and consistency of the imagined results for these unseen combinations using standard generation quality evaluation metrics like FVD (Cobbe et al 2019), FID (Heusel et al 2017) etc.

2. The OOD results contain novel factor combinations. But do they contain entirely unseen object shapes, dynamics etc. ? if no, unless there is a specific challenge prohibiting such an evaluation, these results should also be informative about the model's out of distribution generalisation. For example, generalisation to entirely new object shapes, object textures/colors and object motion could be evaluated.

3. The authors have only evaluated their results on their own datasets. Some results on existing datasets like MOVI or CLEVRER should be very interesting to have and would help contextualise the model's performance. Such an evaluation should provide insights into the challenges of extending the proposed approach to more complex scenes with more diverse objects, motions and occlusions. If there are technical challenge prohibiting such evaluations, the authors should clearly explain what those issues may be.

**Questions:**

see weaknesses.

---

> ### Author Response · Authors · 2024-11-25
>
> *Thank you for this great review and for raising various important points!*
>
> ---
>
> ### R1. Quantitative Results for Future Imagination
>
> > The compositional imagination evaluation only has qualitative results which while interesting is not very informative about the model's performance relative to the other baselines. Some comparative, quantitative results should help here. For example, the authors can holdout a set of combinations in their dataset during training and evaluate the fidelity and consistency of the imagined results for these unseen combinations using standard generation quality evaluation metrics like FVD (Cobbe et al 2019), FID (Heusel et al 2017) etc.
> >
>
> We sincerely appreciate the reviewer's suggestion about quantitative evaluation of compositional imagination. To our knowledge, there is no relevant baseline that can perform compositional imagination in terms of dynamical factors like ours can.
>
> However, we understood the intent of your comment and therefore, we agree about having a metric to measure correctness of the predicted videos. To measure this, we think FID/FVD are not ideal because they only focus on the visual quality. Rather, we think it is better to report the MSE/LPIPS/PSNR of the predicted videos since these metrics also emphasize correctness of the structure and content of the video.
>
> **Compositional Imagination.** We computed MSE, LPIPS, and PSNR to evaluate compositional imagination, where the blocks are manipulated in the final context frame and the remaining video is rolled out. Specifically, for each test video, we performed the following manipulation: we swapped the blocks for each intra-object factor across all object pairs in the video. Alongside, for each of these manipulations, we generated ground truth videos also. To assess the quality of the imagined compositions, we calculated MSE, LPIPS, and PSNR metrics by comparing the compositionally imagined videos to their respective ground truth videos. We have added these results to the paper and we also report them below:
> |  | Moving-Sprites | Dancing-Sprites |
> | --- | --- | --- |
> | MSE | 1261 | 1484 |
> | LPIPS | 0.288 | 0.395 |
> | PSNR | 30.42 | 33.83 |
>
> We would like to mention that to our knowledge, there is no relevant baseline that can do compositional imagination at the level of dynamical factor blocks, and as such, we report these numbers in the hope that these would be useful for future works that may want to compare against our model.
>
> ---
>
> ### R2. Generalization to Entirely Unseen Concepts
>
> > The OOD results contain novel factor combinations. But do they contain entirely unseen object shapes, dynamics, etc.? If not, unless there is a specific challenge prohibiting such an evaluation, these results should also be informative about the model's out of distribution generalization. For example, generalization to entirely new object shapes, object textures/colors and object motion could be evaluated.
> >
>
> Thank you for this insightful suggestion – we sincerely appreciate you raising this important point about evaluating true out-of-distribution generalization.
>
> Motivated by your valuable feedback, we conducted additional experiments on our downstream scene prediction and reasoning task (Section 5.3), specifically incorporating shapes and dance patterns that were completely unseen during the model's pre-training phase. We have carefully documented these new results and their detailed analysis in Appendix Section D.5.
>
> We find that Dreamweaver continues to demonstrate superior performance compared to all baseline approaches, even in this more challenging OOD setting with entirely novel elements. These findings suggest that the block-slot representations learned by Dreamweaver exhibit robust generalization capabilities, enabling effective performance on downstream tasks even when encountering static and dynamic concepts that were absent from the pre-training data.

---

> ### Author Response · Authors · 2024-11-25
>
> ### R3. Evaluation on MOVi or CLEVRER
>
> > The authors have only evaluated their results on their own datasets. Some results on existing datasets like MOVI or CLEVRER should be very interesting to have and would help contextualize the model's performance. Such an evaluation should provide insights into the challenges of extending the proposed approach to more complex scenes with more diverse objects, motions and occlusions. If there are technical challenge prohibiting such evaluations, the authors should clearly explain what those issues may be.
> >
>
> We are deeply grateful for this thoughtful suggestion regarding evaluation on established datasets like MOVi and CLEVRER. We would like to explain several technical considerations that make these datasets less suitable for our current research focus:
>
> - **3D Motion Complexity**: The MOVi dataset involves 3D object motion, which presents unique challenges for our current model architecture. Accurate 3D motion inference from monocular images requires additional information like depth or camera pose data. While MOVi provides this information, incorporating it would significantly expand beyond our current scope of discovering dynamical factor concepts directly from raw pixels.
> - **Factor Diversity**: The MOVi/CLEVRER episodes primarily focus on classical physics with initial velocities as the main dynamical factors. In contrast, our chosen datasets offer a broader spectrum of independent dynamical factors, including movement directions, dance patterns, and appearance changes (e.g., color change patterns)
>
> However, as the reviewer said, we acknowledge that exploring our approach on these established datasets would provide valuable insights. As we are still deliberating regarding these datasets, we will add the above explanation to the paper in the final version.

---

> > ### Comment · Area_Chair_KYjh · 2024-11-30
> >
> > Dear Reviewer,
> >
> > The authors have provided their responses. Could you please review them and share your feedback?
> >
> > Thank you!

---

> > > ### Comment · Reviewer_aS2J · 2024-12-01
> > >
> > > I thank the authors for their reply and for addressing some of my concerns.
> > > Given the absence of results on a more challenging dataset that is not created by the authors, I will maintain my previous score.

---

### Official Review · Reviewer_ixLS · 2024-11-03

**Soundness:** 3
**Presentation:** 2
**Contribution:** 3
**Rating:** 6
**Confidence:** 3

**Summary:**

This paper presents a neural network model named Dreamweaver, designed to learn compositional world representations from videos in an unsupervised manner, without the need for auxiliary data such as text or labeled masks. It utilizes a novel Recurrent Block-Slot Unit (RBSU) to extract modular representations of objects and their attributes, including both static and dynamic attributes. By training to predict future frames rather than reconstructing them, Dreamweaver can generate future video sequences based on learned compositional features and performs exceptionally well in world modeling and compositional reasoning tasks across various datasets.

**Strengths:**

1. Dreamweaver learns compositional world representations without relying on auxiliary data such as text or labeled masks.

2. The proposed RBSU captures both static factors (such as shape) and dynamic factors (such as motion direction), allowing the model to generate new video sequences by recombining learned object attributes.

3. Dreamweaver performs well in new object configurations and arrangements outside the training set, demonstrating strong adaptability.

4. By predicting future frames, Dreamweaver enhances its ability to represent dynamic concepts, outperforming models trained using reconstruction objectives.

**Weaknesses:**

1. The architecture of Dreamweaver relies on complex Recurrent Block Slot Units (RBSUs) and self-regressive Transformer decoders, requiring significant computational resources and memory, especially when processing long video sequences or higher resolution videos.

2. Due to the use of Discrete VAE (dVAE) for image token representation, Dreamweaver may be limited in video generation quality, particularly in applications that require fine visual details. For example, in the Moving-Sprites experiment shown in Figure 4, when objects in the video overlap, the shapes in the generated video frames may become slightly distorted.

3. Although the model can generalize to simple object configurations out of distribution, its generalization ability may be limited in complex scenes. The model's performance on real-world videos with high visual complexity is unknown.

**Questions:**

Although the model can predict short-term dynamic scenes, as the prediction time extends, the generated frames may gradually deviate from the true trajectory. For example, in the Dancing-CLEVR example 3 shown in Figure 4, the last frame's blue sphere is slightly deformed and enlarged. How does the model's performance change as the prediction time increases?

---

> ### Author Response · Authors · 2024-11-25
>
> *We appreciate the reviewer’s insightful observations and constructive feedback!*
>
> ---
>
> ### R1. Computational Demands
>
> > The architecture of Dreamweaver relies on complex Recurrent Block Slot Units (RBSUs) and self-regressive Transformer decoders, requiring significant computational resources and memory, especially when processing long video sequences or higher resolution videos.
> >
>
> We appreciate the reviewer's thoughtful consideration of computational efficiency. While the architectural complexity may appear significant at first glance, we would like to clarify that the computational demands are well-managed through our design choices:
>
> - Dreamweaver processes partial video windows rather than entire sequences, which significantly reduces memory requirements.
> - Additionally, our transformer decoder architecture shares its foundation with STEVE, maintaining comparable computational demands.
>
> To provide concrete evidence, we conducted a detailed comparison of computational costs between our RBSU encoder and STEVE's encoder. This comparison reveals **only a 2.3% increase** in total FLOPs compared to STEVE. The modest increase stems from managing multiple block vectors within slots and recurrent processing, while the CNN backbone remains the primary computational cost in both architectures at 129G FLOPs.
>
> | Model | Network | Number of FLOPs | Number of Parameters |
> | --- | --- | --- | --- |
> | STEVE | Total | 129G | 23.129M |
> |  | CNN Backbone | 129G | 22.159M |
> |  | Recurrent Network | 0.28G | 0.835M |
> | Dreamweaver (DW) | Total | 132G | 30.778M |
> |  | CNN Backbone | 129G | 22.159M |
> |  | Recurrent Network | 3.156G | 8.539M |
>
> We acknowledge the importance of computational efficiency. As a future work, optimizations such as Transformers or State-Space Model-based alternatives to RNN modules for improved resource utilization can be considered.
>
> ---
>
> ### R2. Limitations of dVAE
>
> > Due to the use of Discrete VAE (dVAE) for image token representation, Dreamweaver may be limited in video generation quality, particularly in applications that require fine visual details. For example, in the Moving-Sprites experiment shown in Figure 4, when objects in the video overlap, the shapes in the generated video frames may become slightly distorted.
> >
>
> We sincerely thank the reviewer for this insightful observation! While image quality limitations in scenarios like object overlap are noted, we want to clarify that our paper's primary focus is on advancing compositional representation learning rather than optimizing video generation quality.
>
> In presenting our work, we deliberately chose a minimalistic and transparent implementation to clearly communicate the core framework and its underlying principles to the research community. We believe this approach best demonstrates the fundamental potential of our method.
>
> Since the dVAE is not central to our core contribution, our framework readily accommodates more sophisticated decoding approaches. For instance, the current dVAE could be replaced with advanced architectures like VQ-GAN or diffusion-based models to enhance generation quality while maintaining the key benefits of our compositional learning approach.

---

> ### Author Response · Authors · 2024-11-25
>
> ### R3. Complex and Real-World Scenes
>
> > Although the model can generalize to simple object configurations out of distribution, its generalization ability may be limited in complex scenes. The model's performance on real-world videos with high visual complexity is **unknown**.
> >
>
> We sincerely appreciate the reviewer's valuable feedback regarding real-world applications. We agree that handling real-world videos is an important goal in the field. Our paper's key contribution, however, focuses on a fundamental advancement: the first demonstration of discovering and independently manipulating dynamic compositional factors without relying on language supervision.
>
> The challenge with real-world datasets, despite their rich visual content, lies in the lack of ground truth annotations for independent dynamical factors. This limitation makes it difficult to rigorously validate our core technical claims about factor discovery and compositionality. Our choice of synthetic datasets provides a controlled environment that enables precise quantification of these capabilities.
>
> Our experiments on Moving-CLEVRTex (as demonstrated in Figures 10 and 12) showcase our method's ability to handle substantial visual complexity, including sophisticated textures, lighting effects, and multi-object dynamics. To our knowledge, this dataset represents the highest level of visual complexity achieved in this research direction, matching the complexity standards set by the current state-of-the-art approach [1] on intra-slot disentanglement.
>
> We fully agree that extending these capabilities to real-world scenarios is an important direction for future research. We believe developing benchmarks that combine real-world visual complexity with well-defined dynamic factors would be valuable for the systematic evaluation of compositional video understanding. We look forward to exploring these directions in future work.
>
> [1] Singh, Gautam, Yeongbin Kim, and Sungjin Ahn. "Neural systematic binder."
>
> ---
>
> ### R4. How does the model's performance change as the prediction time increases?
>
> > Although the model can predict short-term dynamic scenes, as the prediction time extends, the generated frames may gradually deviate from the true trajectory. For example, in the Dancing-CLEVR example 3 shown in Figure 4, the last frame's blue sphere is slightly deformed and enlarged. How does the model's performance change as the prediction time increases?
> >
>
> We greatly appreciate the reviewer's astute observation about frame prediction quality over time. While our work's primary focus is on unsupervised discovery of compositional representations rather than long-term prediction accuracy, we agree that addressing these compounding errors represents an important direction for future research in compositional world modeling.
>
> In response, we provide quantitative analyses of prediction error over increasing time horizons, which will serve as a helpful benchmark for future work in this area.
>
> Results on Moving-Sprites:
> |  | $t=1$ | $t=4$ | $t=8$ | $t=12$ |
> | --- | --- | --- | --- | --- |
> | MSE | 6.9761e-11 | 536.9220 | 1854.816 | 3061.3220 |
> | LPIPS | 1.3619e-07 | 0.2773 | 0.3614 | 0.4418 |
> | PSNR | 149.70 | 20.83 | 15.45 | 13.27 |
>
> Results on Dancing-CLEVR:
> |  | $t=1$ | $t=4$ | $t=8$ | $t=12$ |
> | --- | --- | --- | --- | --- |
> | MSE | 8.2764e-11 | 74.7438 | 200.0276 | 280.836 |
> | LPIPS | 4.6022e-07 | 0.0119 | 0.0251 | 0.0312 |
> | PSNR | 148.95 | 29.395 | 25.119 | 23.646 |

---

> > ### Comment · Reviewer_ixLS · 2024-11-26
> >
> > I appreciate the authors' clarification. I will keep the score at 6.

---

### Official Review · Reviewer_aMJV · 2024-11-03

**Soundness:** 2
**Presentation:** 3
**Contribution:** 3
**Rating:** 6
**Confidence:** 4

**Summary:**

This paper proposed a neural architecture called Dreamweaver designed to discover hierarchical and compositional representations from raw videos. The core contribution of this work is  a novel Recurrent Block-Slot Unit (RBSU) to decompose videos into their constituent objects and attributes. Through experiments, the author showed that Dreamweaver can learn a disentangled representation and generate new videos by recombining attributes from different objects.

**Strengths:**

1. Developed a new module Recurrent Block-Slot Unit (RBSU) to decompose videos.

2. Well-written, easy to follow

3. Experimental results show that Dreamweaver can learn different attributes and freely combine them to generate varied videos.

**Weaknesses:**

1. Missing related works:  Some related works [1,2,3] also discuss how to use RNNs for composition video generation or use slot attention to learn disentangled representations. The authors could also include these in the related work section.

2. Insufficient comparison: This paper claim to be the first work that can learn both static and dynamic composable concepts in an unsupervised way. But I think Slotformer and Slotdiffusion[3] can do the same decomposition and are not inclued for comparsion. The authors could additionally supplement this part with experiments or explain why it is unfair to compare with SlotFormer and similar methods.

3. Better datasets: CLEVR and Sprites are still too simple. The authors could experiment on more complex datasets, such as Kubric[4] , to better demonstrate the effectiveness of the method.


Reference:

[1]. Goyal, A., Lamb, A., Hoffmann, J., Sodhani, S., Levine, S., Bengio, Y. and Schölkopf, B., 2019. Recurrent independent mechanisms. arXiv preprint arXiv:1909.10893.

[2]. Yu, W., Chen, W., Yin, S., Easterbrook, S. and Garg, A., 2022. Modular action concept grounding in semantic video prediction. In Proceedings of the IEEE/CVF Conference on Computer Vision and Pattern Recognition (pp. 3605-3614).

[3]. Wu, Z., Hu, J., Lu, W., Gilitschenski, I. and Garg, A., 2023. Slotdiffusion: Object-centric generative modeling with diffusion models. Advances in Neural Information Processing Systems, 36, pp.50932-50958.

[4]. Greff, K., Belletti, F., Beyer, L., Doersch, C., Du, Y., Duckworth, D., Fleet, D.J., Gnanapragasam, D., Golemo, F., Herrmann, C. and Kipf, T., 2022. Kubric: A scalable dataset generator. In Proceedings of the IEEE/CVF conference on computer vision and pattern recognition (pp. 3749-3761).

**Questions:**

All of my question are provided in the Weaknesses section

---

> ### Author Response · Authors · 2024-11-25
>
> *We appreciate the reviewer’s suggestions and for raising important points to consider.*
>
> ---
>
> ### R1. Missing Related Works
>
> > Some related works [1,2,3] also discuss how to use RNNs for composition video generation or use slot attention to learn disentangled representations. The authors could also include these in the related work section.
> >
>
> We sincerely thank the reviewer for pointing out the omissions! We have now added the said papers and a more exhaustive related work section in our revised paper.
>
> ---
>
> ### R2. Insufficient Comparison
>
> > This paper claim to be the first work that can learn both static and dynamic composable concepts in an unsupervised way. But I think Slotformer and Slotdiffusion [3] can do the same decomposition and are not included for comparison. The authors could additionally supplement this part with experiments or explain why it is unfair to compare with SlotFormer and similar methods.
> >
>
> We thank the reviewer for their thoughtful feedback regarding comparisons with SlotFormer and SlotDiffusion. We'd like to clarify the key distinctions that make our work novel:
>
> The important contribution in our work is that Dreamweaver can learn representations of static and dynamic *object attributes* without any supervision. While SlotFormer and SlotDiffusion can decompose scenes into object-level representations, **they learn one monolithic, single vector representation per object**, and **they do not do any intra-object decomposition** like Dreamweaver does.
>
> Furthermore, we do not believe a comparison with SlotFormer and SlotDiffusion is necessary over our current results for the following reasons:
>
> **Comparison with SlotFormer.** Since SlotFormer builds upon pretrained SAVi or STEVE models as its backbone, we believe our existing comparison with STEVE (which has been demonstrated to be the stronger of these two baselines [1]) provides meaningful insight into relative performance. However, to address this point thoroughly, we conducted additional experiments with SAVi across our datasets. The results further support our method's effectiveness:
>
> | Dataset | Metric | Ours | SAVi |
> | --- | --- | --- | --- |
> | Moving Sprites | D | 0.4348 $\pm$ 0.0243 | 0.4093 $\pm$ 0.0490 |
> |  | C | 0.4441 $\pm$ 0.0628 | 0.2321 $\pm$ 0.0376 |
> |  | I | 0.8767 $\pm$ 0.0113 | 0.6131 $\pm$ 0.0036 |
> |  | I-D | 0.8660 $\pm$ 0.0199 | 0.2807 $\pm$ 0.0137 |
> | Dancing CLEVR | D | 0.4825 $\pm$ 0.0507 | 0.3895 $\pm$ 0.075 |
> |  | C | 0.4490 $\pm$ 0.0092 | 0.1879 $\pm$ 0.047 |
> |  | I | 0.8559 $\pm$ 0.0088 | 0.6310 $\pm$ 0.0368 |
> |  | I-D | 0.952 $\pm$ 0.0229 | 0.3547 $\pm$ 0.0082 |
>
> **Comparison with SlotDiffusion.** While SlotDiffusion offers valuable contributions to the field, there are two key considerations that make direct comparison challenging: (1) its focus on object-level rather than intra-object decomposition, and (2) its use of a diffusion decoder versus our transformer-based approach. We believe our comparison with STEVE provides a more appropriate baseline given the architectural similarities. That said, we agree that exploring the integration of diffusion decoders with our approach could be a fascinating direction for future research.
>
> [1] Singh, Gautam, et al. “Simple Unsupervised Object-Centric Learning for Complex and Naturalistic Videos”

---

> > ### Author Response · Authors · 2024-11-25
> >
> > ### R3. Suggestion for testing on Kubric Datasets
> >
> > > Better datasets: CLEVR and Sprites are still too simple. The authors could experiment on more complex datasets, such as Kubric [4], to better demonstrate the effectiveness of the method.
> > >
> >
> > We appreciate the reviewer for this valuable suggestion regarding dataset complexity. We'd like to address this thoughtful point in two parts:
> >
> > **Current Dataset Complexity.** Our Moving-CLEVRTex dataset represents a significant level of visual complexity, comparable to current state-of-the-art work in intra-object disentanglement [1]. While recent works like SAVi++ and DINOSAUR [2, 3] have indeed shown impressive results on realistic scenes like Kubric, it's important to note that these methods focus on object-level decomposition rather than the more granular intra-object factor disentanglement that our work addresses. As the first method to discover dynamical intra-object factor concepts directly from raw pixels, we believe our current results establish an important foundation for future work on more complex scenes.
> >
> > **Considerations Regarding Kubric/MOVi Datasets.** While we appreciate the suggestion to use Kubric, there are several technical considerations that make it less suitable for our current research focus:
> >
> > - **3D Motion Complexity**: The MOVi dataset involves 3D object motion, which presents unique challenges for our current model architecture. Accurate 3D motion inference from monocular images requires additional information like depth or camera pose data. While MOVi provides this information, incorporating it would significantly expand beyond our current scope of discovering dynamical factor concepts directly from raw pixels.
> > - **Factor Diversity**: The MOVi episodes primarily focus on classical physics with initial positions and velocities as the main dynamical factors. In contrast, our chosen datasets offer a broader spectrum of independent dynamical factors, including movement directions, dance patterns, and appearance changes (e.g., color change patterns).
> >
> > [1] Singh, Gautam, Yeongbin Kim, and Sungjin Ahn. "Neural systematic binder."
> >
> > [2] Elsayed, Gamaleldin, et al. "Savi++: Towards end-to-end object-centric learning from real-world videos."
> >
> > [3] Seitzer, Maximilian, et al. "Bridging the gap to real-world object-centric learning."

---

> > > ### Comment · Reviewer_aMJV · 2024-11-26
> > > **Reply to rebuttal**
> > >
> > > I appreciate the authors' clarification. They address my main concerns and questions about this paper. I will raise score to 6.

---

### Author Response · Authors · 2024-11-25
**General Response to All Reviewers and ACs**

We sincerely thank all the reviewers and Area Chairs for reviewing our work and providing thoughtful, insightful feedback. We are pleased that the reviewers found several strengths in our work, as highlighted below:

- **Well-written and easy to follow** (Reviewers aMJV and ZQtq)
- **Novelty in the proposed core module**, Recurrent Block-Slot Units, effectively capturing both static and dynamic interpretable concepts (Reviewers aMJV, ixLS, and aS2J)
- **Compositional world representations** achieved without auxiliary data (Reviewer ixLS)
- **Generalization ability** to out-of-distribution block configurations (Reviewers aS2J and ixLS)


**Revision Summary**

In response to the reviewers' comments and suggestions,

- We updated **Section 4** and **Appendix A** to provide further comprehensive related works as Reviewer **aMJV** suggested.
- We added **Figure 13** and a small revision in **Appendix E** to provide a clearer view of our designed datasets in response to Reviewers **aMJV** and **ZQtq**.
- We shared a quantitative performance of compositional imagination results (**Appendix D4**) as Reviewer **aS2J** suggested.
- We added additional experimental results in **Appendix D5** on the generalization ability of our model to entirely unseen concepts (novel shapes and dynamics) as Reviewer **aS2J** suggested.
    - Interestingly, we found our model shows only a slight drop in performance for the downstream reasoning task even when encountering shapes or dynamics that were not seen in the pre-training phase. For the details, see **Figure 12**.

---

### Meta-Review · Area_Chair_KYjh · 2024-12-14

**Metareview:**

The paper introduces Dreamweaver, a neural architecture leveraging a novel Recurrent Block-Slot Unit (RBSU) for hierarchical video decomposition and multi-frame prediction. It outperforms state-of-the-art baselines in world modeling and enables compositional imagination through modularized concept representations.

The manuscript is well-written, with a well-motivated and justified method design. The proposed module is novel and offers valuable contributions to video generation models.

The paper received unanimous acceptance with scores of 6. However, reviewers initially raised concerns regarding missing comparisons with baselines, limited benchmarks on complex datasets, and insufficient analysis of the model's OOD generalization. During the rebuttal, the authors conducted extensive experiments addressing these concerns, which clarified the reviewers’ doubts.

Given the method's novelty and its superior performance over other baselines, the AC recommended acceptance, strongly encouraging the authors to incorporate the rebuttal responses into the final version.

**Additional Comments On Reviewer Discussion:**

The paper received unanimous acceptance with scores of 6. However, reviewers initially raised concerns regarding missing comparisons with baselines, limited benchmarks on complex datasets, and insufficient analysis of the model's OOD generalization. During the rebuttal, the authors conducted extensive experiments addressing these concerns, which clarified the reviewers’ doubts.

Given the method's novelty and its superior performance over other baselines, the AC recommended acceptance, strongly encouraging the authors to incorporate the rebuttal responses into the final version.

---

### Decision · Program_Chairs · 2025-01-22

Accept (Poster)